# Spatially aware dimension reduction for spatial transcriptomics

**Lulu Shang** [1,2] **& Xiang Zhou** [1,2] ✉

Spatial transcriptomics are a collection of genomic technologies that have enabled transcriptomic profiling on tissues with spatial localization information. Analyzing spatial transcriptomic data is computationally challenging, as the data collected from various spatial transcriptomic technologies are often noisy and display substantial spatial correlation across tissue locations. Here, we develop a spatially-aware dimension reduction method, SpatialPCA, that can extract a low dimensional representation of the spatial transcriptomics data with biological signal and preserved spatial correlation structure, thus unlocking many existing computational tools previously developed in single-cell RNAseq studies for tailored analysis of spatial transcriptomics. We illustrate the benefits of SpatialPCA for spatial domain detection and explores its utility for trajectory inference on the tissue and for high-resolution spatial map construction. In the real data applications, SpatialPCA identifies key molecular and immunological signatures in a detected tumor surrounding micro-environment, including a tertiary lymphoid structure that shapes the gradual transcriptomic transition during tumorigenesis and metastasis. In addition, SpatialPCA detects the past neuronal developmental history that underlies the current transcriptomic landscape across tissue locations in the cortex.

Spatial transcriptomics is a collection of groundbreaking new genomics technologies that enable the measurement of gene expression with spatial localization information on tissues. Specifically, the next-generation DNA sequencing-based technologies represented by Slide-seq[1,2], hybridization-based technologies represented by STARmap[3], and spatial transcriptomics through spatial barcoding from 10x genomics[4], can measure tens of thousands of genes on thousands of tissue locations, each of which consisting of a few to a few dozen single cells. In situ RNA sequencing-based technologies, such as targeted in situ sequencing (ISS)[5] and FISSEQ[6], can measure the entire transcriptome at a single-cell resolution. The single-molecular fluorescence in situ hybridization (smFISH)-based technologies, represented by MERFISH[7–9], seqFISH[10,11], and seqFISH plus[12], can measure hundreds to tens of thousands of genes on subcellular organelles inside single cells across the tissue. These technologies altogether have enabled the study of the spatial transcriptomic landscape of tissues, catalyzing new discoveries in many areas of biology[13,14].

Regardless of which technology one uses, the expression measurements obtained from spatial transcriptomics, just like any other genomic data types, are often noisy[15]. A common data processing step for extracting informative signals from noisy data in other genomic data types is dimension reduction. Dimension reduction aims to enrich biological signals through inferring a low-dimensional representation of the original genomic data. Dimension reduction is commonly applied to many genomics studies, including the recent single-cell RNA sequencing (scRNA-seq) studies[16]. In scRNA-seq, dimension reduction has become an indispensable data processing step for noise removal, facilitating data visualization and multiple downstream analyses that include cell-type clustering[17] and lineage inference[18,19]. Many dimension reduction methods have been previously developed for scRNA-seq[20–22] and some of these approaches have been directly applied to spatial transcriptomics. For example, Seurat[23] recommends the use of principal component analysis (PCA) to preprocess spatial transcriptomics data. STUtility[24] performs dimension reduction in spatial

[1]Department of Biostatistics, University of Michigan, Ann Arbor, MI 48109, USA. [2]Center for Statistical Genetics, University of Michigan, Ann Arbor, MI 48109, USA. ✉e-mail: xzhousph@umich.edu

transcriptomics using non-negative matrix factorization (NMF), where the NMF factors are further interpreted through association analysis with distinct gene pathways. Despite these initial applications, however, scRNA-seq dimension reduction methods are not tailored for spatial transcriptomics and are not fully effective there. In particular, scRNA-seq-based dimension reduction methods do not make use of the rich localization information contained in spatial transcriptomics and are not able to take advantage of the spatial correlation structure across tissue locations. Intuitively, in spatial transcriptomics, the neighboring locations on a tissue often share similar composition of cell types and display similar gene expression levels. Consequently, the low-dimensional components of neighboring locations are likely to be similar to each other, more so than those on locations that are far away. Accounting for the similarity in the low-dimensional components on neighboring locations could facilitate effective dimension reduction in spatial transcriptomics and enable tailored downstream analyses.

In this study, we explore the benefits of dimension reduction for a particular analytic task in spatial transcriptomics that is commonly referred to as spatial domain detection. Spatial domains represent spatially organized and functionally distinct anatomical structures on the tissue that are each characterized by unique local features with varying cell-type composition, transcriptome heterogeneity, and cell-cell interactions[25-27]. Detecting spatial domains on the tissue is a critical first step towards understanding how these domains are coordinated with each other in carrying out the tissue functions and how they are generated through the complex developmental process. Several methods have been recently developed for detecting spatial domains in spatial transcriptomics, each with its own advantages and drawbacks[28-32]. For example, BayesSpace[31] detects spatial domains in spatial transcriptomics (ST) or 10x Visium data through explicit modeling of the specific spatial arrangement of the measured locations on the tissue. In the process of domain detection, BayesSpace can also enhance the spatial transcriptomics data with a fixed resolution, though it comes with a relatively heavy computational burden. As another example, SpaGCN[28] detects spatial domains using a graph convolutional network, which relies on an adjacency matrix to incorporate spatial and histological information in the graph convolution layer. Despite its computational efficiency, as we will show here, SpaGCN is effective primarily in the setting where each spatial domain is dominated by one or two cell types. As a third example, stLearn[29] extracts morphological features from an H&E image that accompanies certain spatial transcriptomics technologies to perform spatial smoothing on the expression data, with which it performs spatial domain detection and further trajectory inference on pairs of spatial domains. However, the applications of stLearn are limited to spatial transcriptomics that collect H&E images. In the absence of H&E image, stLearn software directly uses PCA and Louvain clustering for spatial domain detection. As a fourth example, HMRF builds a graph to represent the spatial relationship among cells and detects spatial domains by comparing the gene expression of each cell with its surroundings to search for coherent spatial patterns.

In this work, we develop a spatially aware dimension reduction method, which we refer to as the spatial probabilistic PCA, or SpatialPCA. SpatialPCA enables tailored dimension reduction in spatial transcriptomics and facilitates effective downstream analyses that include spatial domain detection. A key feature of SpatialPCA is its ability to explicitly model the spatial correlation structure across tissue locations, thus preserving the neighboring similarity of the original data in the low-dimensional manifold. The low-dimensional components obtained from SpatialPCA contain valuable spatial correlation information and can be directly paired with existing computational tools developed in scRNA-seq for effective and improved downstream analyses in spatial transcriptomics. In particular, the low-dimensional components from SpatialPCA can be paired with scRNA-seq clustering methods to enable effective de novo tissue domain detection and can

be paired with scRNA-seq trajectory inference methods to enable effective developmental trajectory inference on the tissue. Because of the data-generative nature of SpatialPCA and its explicit modeling of spatial correlation, it can also be used to impute the low-dimensional components on new and unmeasured arbitrary spatial locations, facilitating the construction of a refined spatial map with a resolution much higher than that measured in the original study. We illustrate the benefits of SpatialPCA through comprehensive simulations and applied it on four spatial transcriptomics datasets obtained with distinct technologies and tissue structures.

## Results
### Method overview
SpatialPCA is described in "Methods", with its technical details provided in Supplementary Text and its method schematic displayed in Fig. 1a. Briefly, SpatialPCA is a spatially aware dimension reduction method that aims to infer a low-dimensional representation of the gene expression data for spatial transcriptomics. SpatialPCA builds upon the probabilistic version of PCA, incorporates localization information as additional input, and uses a kernel matrix to explicitly model the spatial correlation structure across tissue locations. Because the inferred low-dimensional components from SpatialPCA contain valuable spatial correlation information, we refer to these inferred components as spatial principal components, or spatial PCs. The spatial PCs can be paired with various analytic tools already developed in scRNA-seq studies to enable effective and improved downstream analyses for spatial transcriptomics. We illustrate the benefits of SpatialPCA through four different downstream analyses: spatial transcriptomics visualization, spatial domain detection, spatial trajectory inference on the tissue, and high-resolution spatial map reconstruction. SpatialPCA is implemented as an R package, freely available at www.xzlab.org/software.html.

### Simulations
We performed comprehensive and realistic simulations to evaluate the performance of SpatialPCA and compare it with several other methods (Fig. 1b). The compared methods in simulations include three spatial domain detection methods (BayesSpace[31], SpaGCN[28], and HMRF[33]) and two dimension reduction methods previously used for spatial transcriptomics (PCA[23,34] and non-negative matrix factorization, NMF[24]). The simulation details are provided in "Methods". Briefly, we obtained the cortex tissue from the DLPFC data, segmented it into four cortical layers, specified a distinct cell-type composition of four cell types on each layer, and obtained 10,000 single-cell locations on the tissue. In parallel, we simulated expression counts for 5000 genes and 10,000 single cells from four cell types based on a separate scRNA-seq data. We then assigned the simulated cells onto the single-cell locations of the cortex based on the specified cell-type composition in each cortical layer to create spatial transcriptomics data at single-cell resolution. With the single-cell resolution spatial transcriptomics, we created additional spot-level spatial transcriptomics data at different regional resolutions by merging the expression measurements of single cells into pre-defined spots on the tissue. The simulated spatial transcriptomics resemble the real data and share similar mean–variance relationships across genes (Supplementary Fig. 1a). For each simulated spatial transcriptomics, we then applied SpatialPCA and the other methods to perform spatial domain detection. We evaluated accuracy of different methods by comparing the detected spatial domains with the four underlying cortical layers that are served as the ground truth. We considered four simulation scenarios that cover a range of cell-type compositions, with three simulation settings per scenario in the spot-level resolution spatial transcriptomics. We performed ten simulation replicates for each setting.

In the simulated single-cell resolution spatial transcriptomics, SpatialPCA outperforms the other methods for tissue domain

detection. Specifically, the median adjusted Rand index (ARI) by SpatialPCA across simulation replicates is 0.942, 0.877, 0.931, and 0.439 for the four scenarios, respectively (Fig. 1b and Supplementary Table 1). HMRF and SpaGCN work well in scenario 1 when there is only one dominant cell type in each spatial domain, but its performance

decays in other scenarios when multiple dominant cell types are present in each spatial domain (HMRF median ARI = 0.773, 0.279, 0.617, 0.002; SpaGCN median ARI = 0.625, 0.277, 0.412, 0.138). BayesSpace does not perform as well as SpatialPCA, HMRF or SpaGCN (median ARI = 0.367, 0.225, 0.286, 0.075), presumably because BayesSpace is

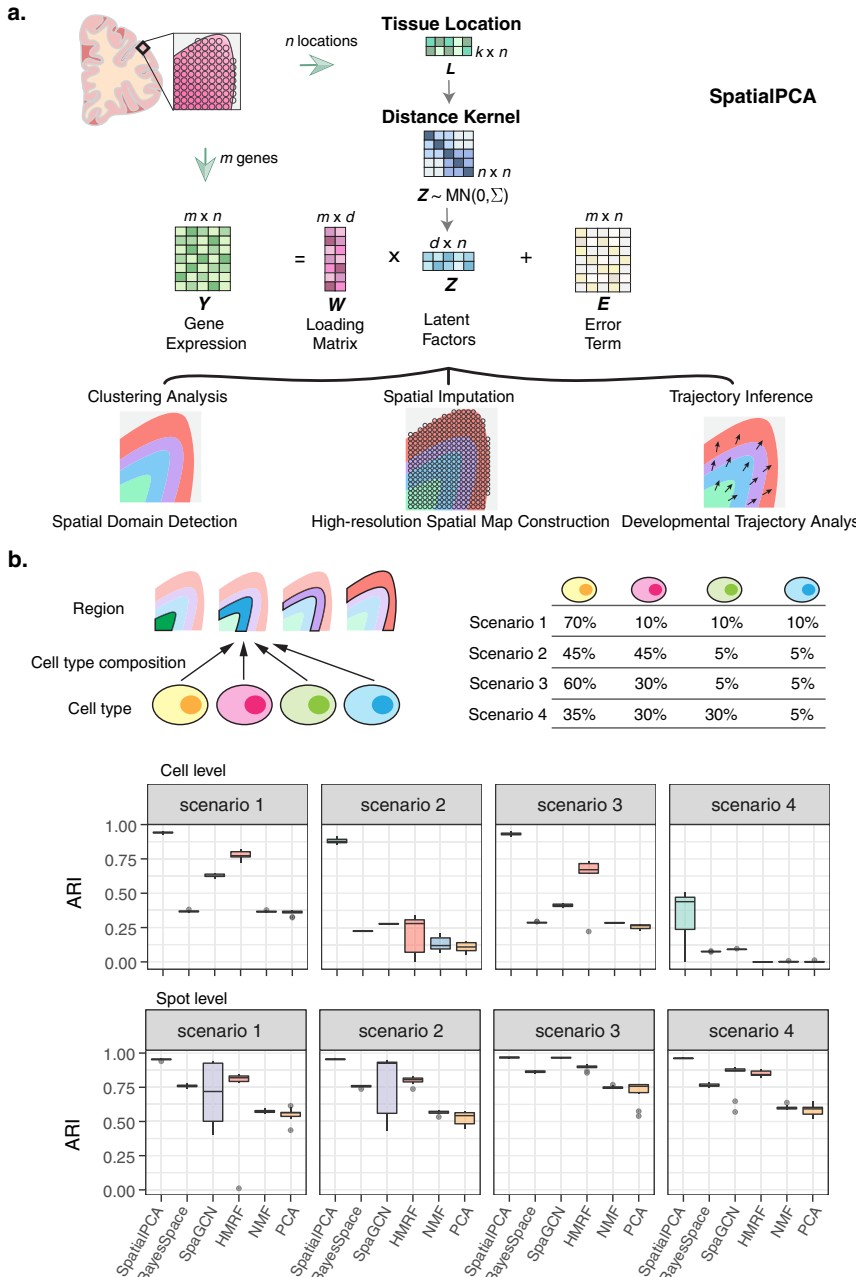

**Fig. 1 | Method schematic of SpatialPCA and simulation results. a** SpatialPCA is a spatially aware dimension reduction method that takes both gene expression $\boldsymbol{Y}$ and location information $\boldsymbol{L}$ as inputs. It models the gene expression matrix $\boldsymbol{Y}$ as a function of latent factors $\boldsymbol{Z}$ through a factor analysis model. Importantly, SpatialPCA builds a kernel matrix $\boldsymbol{\Sigma}$ using the location information $\boldsymbol{L}$ to explicitly model the spatial correlation structure in the latent factors $\boldsymbol{Z}$ across tissue locations. Consequently, the inferred low-dimensional components $\boldsymbol{Z}$ from SpatialPCA contain valuable spatial correlation information and can be paired with various analytic tools already developed in scRNA-seq studies to enable effective and improved downstream analyses for spatial transcriptomics. The examined downstream analyses of spatial transcriptomics include spatial transcriptomics visualization, spatial domain detection, trajectory inference on the tissue, and high-resolution spatial

map construction. **b** In simulation, we obtained cortex tissue from the DLPFC data and manually segmented it into four cortical layers. We specified a distinct cell-type composition for each cortical layer and simulate four scenarios. Then we assigned the simulated cells to the locations of the cortex based on the specified cell-type composition in each layer to create the spatial transcriptomics data. We simulated spatial transcriptomics data both at single-cell resolution (sample size $n = 10{,}000$ cells) and spot-level ($n = 5077$ spots) data. We use adjusted Rand index (ARI) to measure the spatial clustering accuracy at single-cell resolution and spot level (spot diameter is 90 μm). The higher ARI indicates better spatial clustering performance. SpatialPCA outperforms the other methods for detecting the spatial domains in the simulations. In the boxplot, the center line, box limits and whiskers denote the median, upper, and lower quartiles, and 1.5× interquartile range, respectively.

specifically designed for 10X ST and Visium data, it detects neighboring spots based on array coordinates and thus may not be well suited for analyzing single-cell resolution spatial transcriptomics. PCA and NMF perform similarly as BayesSpace, which is not surprising as BayesSpace does not make use of spatial information in single-cell resolution data and effectively becomes PCA there. The ability of SpatialPCA in clustering cells into tissue domains is in direct contrast to PCA and NMF (Supplementary Fig. 1b), which are effective in clustering cells into cell types (Supplementary Fig. 1c). Indeed, methods that accurately cluster cells into spatial domains (e.g., SpatialPCA, HMRF, and SpaGCN) generally do not cluster cells into cell types well, and vice versa, highlighting the distinct goals of cell-type clustering and spatial domain detection. Besides accurately detecting spatial domains, the detected spatial domains from SpatialPCA are spatially continuous and smooth, indicated by low local inverse Simpson index (LISI) score, percentage of abnormal spots (PAS) score, and spatial chaos score (CHAOS), more so than the other methods in all scenarios (Supplementary Fig. 2). Finally, we note that while each domain detection method by default uses a different set of genes as input, all methods work almost equally well when a different set of genes are used as input (Supplementary Fig. 1b).

In the simulated spot-level spatial transcriptomics, SpatialPCA also outperforms the other methods in the majority of simulation settings (Fig. 1b, Supplementary Fig. 3, and Supplementary Tables 2 and 3). For example, in the simulation setting with spot diameter being 90 μm, the median ARI for SpatialPCA is 0.955, 0.953, 0.969, and 0.962 for the four scenarios, respectively (Fig. 1b). HMRF performs well in scenario 3 with two dominant cell types and a small proportion of minor cell types (median ARI = 0.82, 0.806, 0.898, and 0.843). SpaGCN performs well in scenario 2 and 3 with a small proportion of minor cell types in each spatial domain but does not fare well in scenarios 1 and 4 where a larger proportion of minor cell types is present (median ARI = 0.719, 0.925, 0.966, and 0.874). BayesSpace performs slightly better than SpaGCN in scenario 1 but worse in scenarios 2–4 (median ARI = 0.759, 0.759, 0.864, 0.763). PCA and NMF, on the other hand, do not perform well in all scenarios. Besides accurately detecting spatial domains, the detected spatial domains from both SpatialPCA and BayesSpace are continuous and smooth, indicated by low CHAOS (Supplementary Table 4) and PAS (Supplementary Table 5) scores, more so than the other methods in all settings.

Besides the above main simulations, we examined three additional simulation scenarios where we introduced a stripe pattern on the tissue to examine the performance of SpatialPCA and the other methods in detecting spatially non-contiguous and non-smoothly varying domains (details in Method). The results are largely consistent with the main simulations with SpatialPCA outperforming the other methods in all three scenarios (Supplementary Fig. 4). We also explored an alternative simulation strategy where we introduced additional spatial correlation between spots by splitting the expression of a proportion of cells to be randomly added to the expression of its neighboring cells (details in "Methods"). The results are also largely consistent with the main simulations with SpatialPCA outperforming the others (Supplementary Fig. 5).

An important benefit of SpatialPCA is its ability to impute the low-dimensional components on new, unmeasured tissue locations, thus leading to a refined spatial map with a resolution much higher than that of the original study (details in Method). We compared high-resolution spatial map clustering results for spot-level simulation in four scenarios at spot diameter being 145 μm. We compared SpatialPCA with BayesSpace for the clustering results on the imputed subspots, SpatialPCA (median ARI = 0.901, 0.851, 0.873, and 0.706) outperforms BayesSpace (median ARI = 0.534, 0.435, 0.463, and 0.123) in all four scenarios. Besides accurately reconstructing high-resolution maps, the imputed spatial domains from SpatialPCA is more continuous and smoother than BayesSpace, indicated by low CHAOS and

PAS scores in all settings (Supplementary Fig. 6a). SpatialPCA also accurately predicts gene expression on the unmeasured locations. SpatialPCA has higher Pearson's correlation (median correlation = 0.109, 0.11, 0.109, and 0.107) between the predicted gene expression with true expression than BayesSpace (median correlation = 0.085, 0.085, 0.085, and 0.082) on the imputed subspots in all four scenarios (Supplementary Fig. 6b).

## Human dorsolateral prefrontal cortex data by Visium

We applied SpatialPCA and the other methods to analyze four published datasets obtained using different spatial transcriptomics technologies ("Methods"). The four datasets include a human dorsolateral prefrontal cortex data generated by Visium from 10x Genomics, a cerebellum data generated by Slide-seq, and hippocampus data generated from Slide-seq V2, HER2-positive breast tumor data generated by spatial transcriptomics from 10x Genomics. Besides SpatialPCA, we also examined the performance of BayesSpace, SpaGCN, HMRF, stLearn, PCA, and NMF for spatial domain detection, and stLearn, PCA and NMF for trajectory inference. Among these methods, we were unable to apply BayesSpace and HMRF to the Slide-seq V2 data due to the large data size and heavy computational burden. In addition, we only applied stLearn to DLPFC due to a lack of H&E image in the Slide-seq and Slide-seq V2 data, and a lack of JSON file in the HER2-positive breast tumor data.

First, we analyzed the human dorsolateral prefrontal cortex data[35] (Fig. 2), which contains twelve samples with an average of 3973 spots. We used sample 151676 as a main example which contains expression measurement of 33,538 genes on 3460 spots. We first performed dimension reduction on the expression data using either SpatialPCA, PCA or NMF. For each method in turn, we summarized the inferred low-dimensional components into three UMAP or tSNE components and visualized the three resulting components with red/green/blue (RGB) colors in the RGB plot (Fig. 2b). The resulting RGB plots are not sensitive to the scaling of the input data (Supplementary Fig. 7a–f). The RGB plot from SpatialPCA displays the laminar organization of the cortex and shows smooth color transition across neighboring spots and neighboring spatial domains (Supplementary Fig. 7g, h). We evaluated the predictive ability of the output from each method in predicting the true spatial domains using the McFadden-adjusted pseudo-$R^2$ [36] (details in "Methods"). We found that the spatial PCs are highly predictive of the cortical structures annotated by experts based on cytoarchitecture in the original study (pseudo $R^2$ = 0.89). In contrast, the low-dimensional components from PCA (pseudo $R^2$ = 0.751) and NMF (pseudo $R^2$ = 0.633) are less predictive of the known cortical structures as compared to the spatial PCs (Fig. 2c). In addition, we found that the spatial PC associated genes are enriched in synapse-related, neuron projection, and synaptic signaling pathways (Supplementary Fig. 8 and Supplementary Table 6). Clustering based on the spatial PCs identified seven spatial domains that correspond to the annotated cortical layers 1 through 6 and white matter (Fig. 2a). Such clustering results are robust with respect to the number of spatial genes, the number of spatial PCs, the number of clusters specified, the kernel matrix used, and the bandwidth selected for modeling spatial correlation, as well as the clustering methods (Supplementary Figs. 9 and 10). The identified layers by SpatialPCA are enriched with known layer marker genes, including *CXCL14* (layer 2), *SV2C* (layer 3), *HTR2C* (layer 5), and *NR4A2* (layer 6/6b)[37] (Fig. 2d), and depict the annotated spatial domains more accurately (median ARI = 0.542) than Bayes-Space (ARI: median = 0.438), SpaGCN (ARI: median = 0.443), HMRF (ARI: median = 0.304), stLearn (ARI: median = 0.470), PCA (ARI: median = 0.358) and NMF (ARI: median = 0.262; Fig. 2e, Supplementary Figs. 11, 12, and Supplementary Table 7). In addition, the spatial domains detected by both SpatialPCA (median LISI = 1.057; median CHAOS = 0.059; median PAS = 0.02) and BayesSpace (median LISI = 1.088; median CHAOS = 0.061; median PAS = 0.032) are spatially

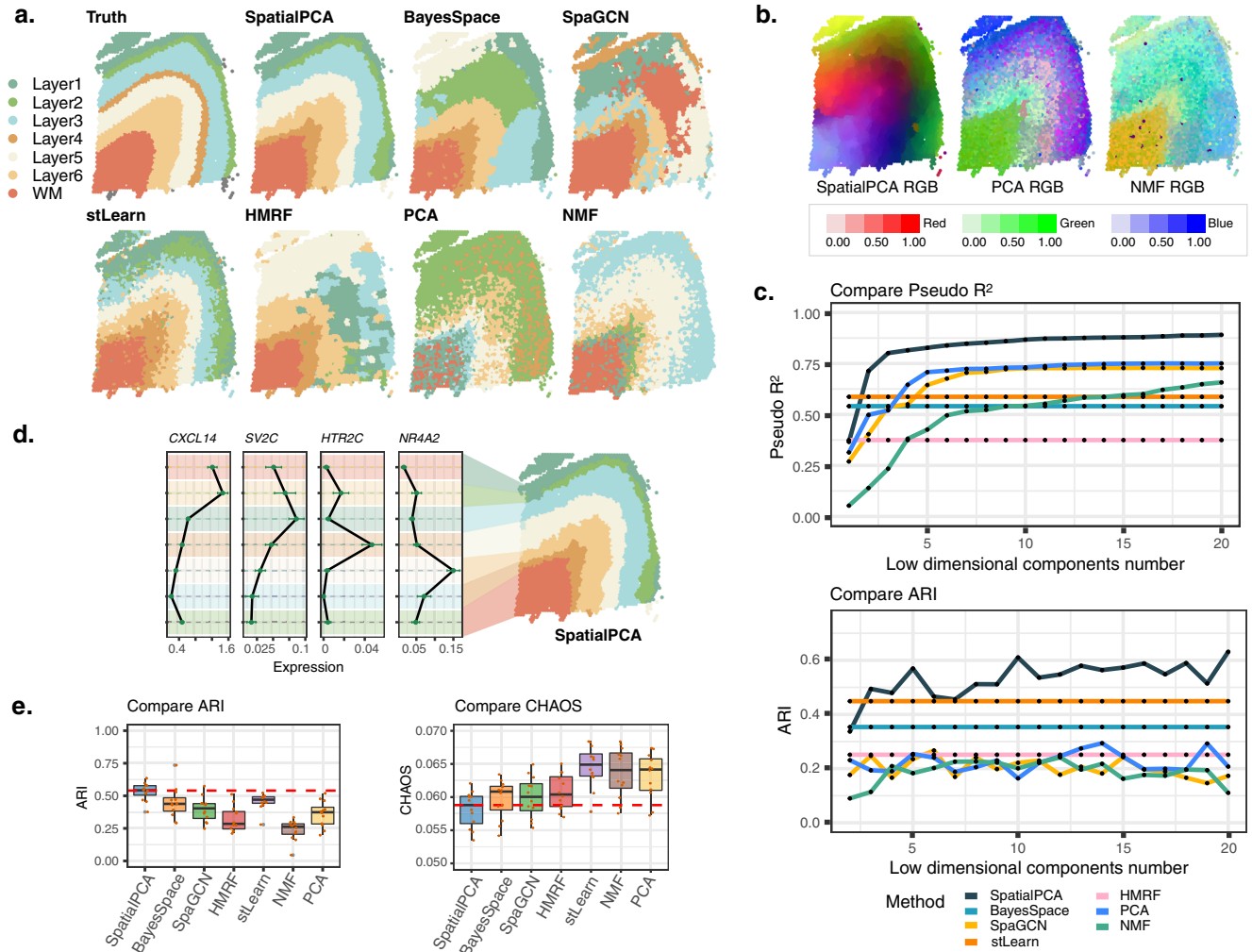

**Fig. 2 | Analysis of the cortex data from DLPFC. a** Clustering of tissue locations based on SpatialPCA, BayesSpace, SpaGCN, stLearn, HMRF, PCA, and NMF. Ground truth of tissue regions on sample 151676 of the human prefrontal cortex are annotated by the original DLPFC study. **b** For SpatialPCA, PCA, and NMF, we summarized the inferred low-dimensional components into three UMAP components and visualized the three resulting components with red/green/blue (RGB) colors through the RGB plot. Color code corresponds to the RGB values of each location's three UMAP components inferred from low-dimensional components in dimension reduction. Different colors indicate different values for each of the three UMAP components on the tissue section, highlighting the difference of the low-dimensional components from different methods included in the panel. **c** Upper panel: spatial PCs in SpatialPCA have higher prediction accuracy for the ground truth of tissue regions in terms of McFadden's pseudo-$R^2$ than latent components from PCA, NMF, and SpaGCN. For BayesSpace and HMRF, we treated their inferred cluster labels as the predictors. Lower panel: spatial PCs in SpatialPCA have higher clustering accuracy for the ground truth of tissue regions in terms of ARI.

**d** Mean expression of layer-specific markers including layer 2 marker gene *CXCL14*, layer 3 marker gene *SV2C*, layer 5 marker gene *HTR2C*, and layer 6/6b marker gene *NR4A2* ($n = 3460$ spots). The cluster labels correspond to the labels of SpatialPCA detected spatial domains in (**a**). In the boxplot, the center line denotes the mean value of the expression. **e** Left: clustering accuracy of different methods in recapitulating the true tissue domains. Accuracy is measured by the adjusted Rand index (ARI) in all 12 sections. For BayesSpace, SpaGCN, and HMRF, clustering was performed based on their default settings. For dimension reduction methods (PCA and NMF), clustering was performed based on the inferred low-dimensional components on spatially variable genes. Right: clustering performance of different methods in obtaining smooth and continuous spatial domains measured by spatial chaos score (CHAOS) in all 12 sections. Lower CHAOS score indicates better spatial continuity of the detected spatial domains. In the boxplot, the center line, box limits and whiskers denote the median, upper, and lower quartiles, and 1.5× inter-quartile range, respectively.

continuous and smooth, more so than that detected by the other methods (HMRF: median LISI = 1.138, CHAOS = 0.06, PAS = 0.049; SpaGCN: median LISI = 1.234, CHAOS = 0.06, PAS = 0.111; stLearn: LISI = 1.455, CHAOS = 0.065, PAS = 0.175; PCA: LISI = 1.591, CHAOS = 0.064, PAS = 0.246; and NMF: LISI = 1.449, CHAOS = 0.064, PAS = 0.207; Fig. 2e and Supplementary Fig. 11). We performed differential gene expression (DE) analysis to identify regional-specific genes (Supplementary Table 8) and found them to be highly enriched in the pathways of myelin sheath, neurogenesis, and neuron projection (Supplementary Fig. 13). The enriched pathways are critical for information processing and neural development in the cortex layers[38]. SpatialPCA can be easily extended to make use of histology

information as the features extracted from H&E images can be incorporated as pseudo-dimension in the distance matrix. However, the histology feature vectors are not highly predictive for the pathologist annotations (median pseudo $R^2 = 0.220$ across 12 tissue sections). Consequently, when histological feature extracted from SpaGCN (Supplementary Fig. 14a) were included in SpatialPCA, we observed a slight decrease in spatial clustering performance (median ARI = 0.539, Supplementary Fig. 14b).

We performed trajectory inference using the spatial PCs and detected one trajectory on the tissue (Supplementary Fig. 15a). The detected trajectory projects from inner to outer layers and captures the well-known "inside-out" pattern of corticogenesis: new neurons are

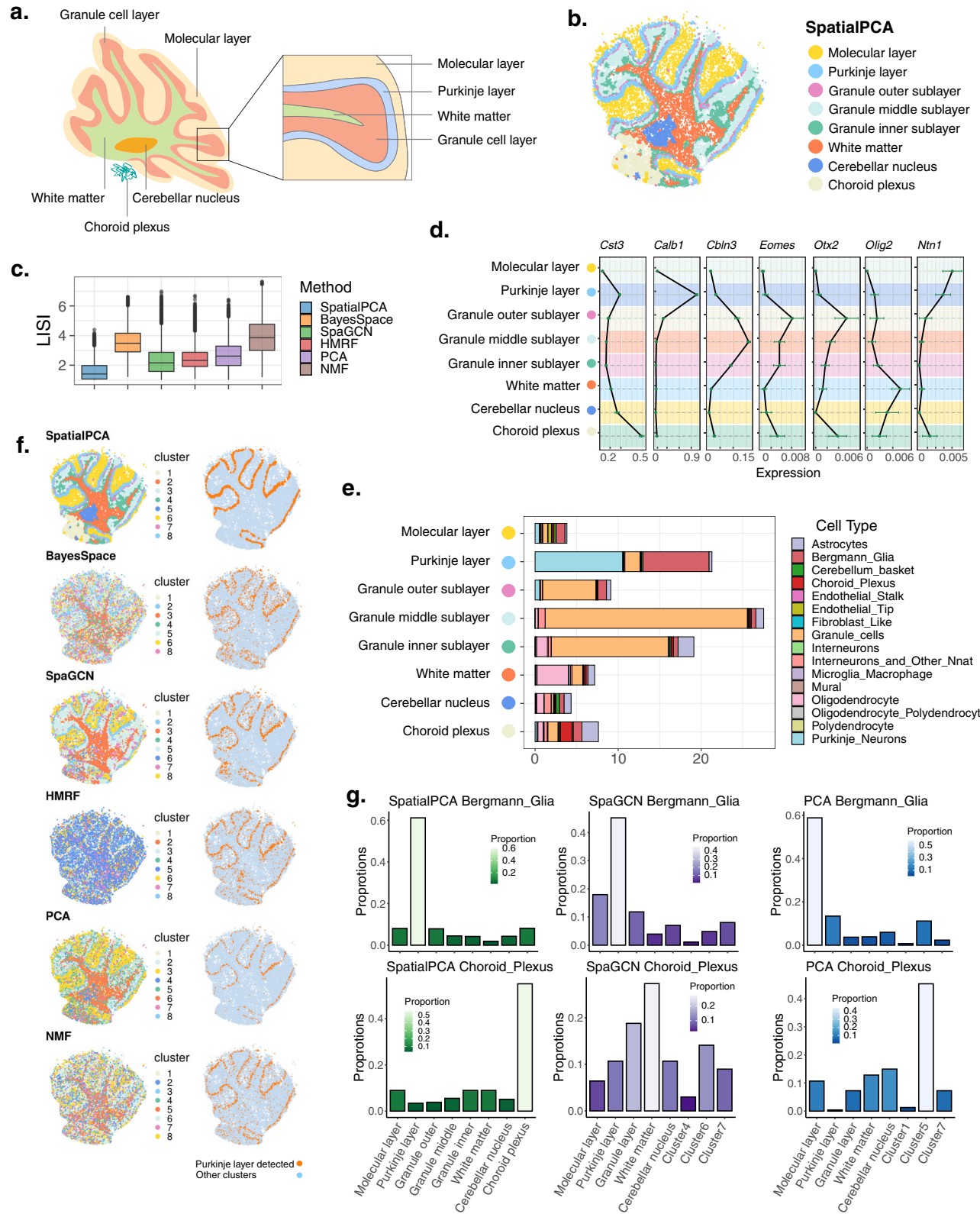

born in the ventricular zone, migrate along the radial glia fibers in a vertical fashion towards the marginal zone on the outskirt of the cortex, and pass the old neurons in the existing layers to form the new cortical layers[39,40]. In contrast, the trajectories inferred based on the low-dimensional components of PCA and NMF point towards almost random directions on the tissue (Supplementary Fig. 15b, c). We also performed trajectory inference using stLearn, which detected multiple

trajectories projecting from the white matter region to different cortical layers (Supplementary Fig. 15d, e) with spot-level trajectories oriented towards almost random directions on the tissue. We further examined genes that are associated with the inferred pseudo-time (Supplementary Fig. 16a and Supplementary Table 9) and found that the pseudo-time associated genes by SpatialPCA are highly enriched in synapse-related, synapse signaling and neuron projection pathways.

**Fig. 3 | Analysis of the cerebellum data from Slide-seq. a** The structure of the mouse cerebellum cortex, with the main tissue regions annotated. **b** Clustering on the low-dimensional components inferred by SpatialPCA segregates tissue locations into distinct tissue regions. The detected tissue regions were annotated based on their relative positions on the tissue and the enriched cell types in each detected tissue domain. **c** Clustering performance of different methods in obtaining smooth and continuous spatial domains measured by local inverse Simpson's index (LISI) in 20,982 locations. Lower LISI score indicates more homogeneous neighborhood spatial domain clusters of a spot. In the boxplot, the center line, box limits and whiskers denote the median, upper and lower quartiles, and 1.5× interquartile range, respectively. **d** Mean expression of regional marker genes in the cerebellum (*n* = 20,982 locations). The cluster labels correspond to the labels of SpatialPCA

regions in (**b**). In the boxplot, the center line denotes the mean value of the expression. **e** Percentage of different cell types (*x* axis) in each tissue domain detected by SpatialPCA (*y* axis). **f** SpatialPCA correctly depicts the Purkinje layer. Left: tissue location clustering results using different methods; different color represents different location clusters. Right: the Purkinje layer detected by different methods is highlighted in orange, and the background is colored in light blue. For spatial clustering methods (BayesSpace, SpaGCN, and HMRF), location clusters were inferred using the software. For dimension reduction methods (PCA and NMF), clustering was performed based on the inferred low-dimensional components after dimension reduction using spatially variable genes. **g** Distribution of Bergmann glia cells and choroid plexus cells in each cluster for different methods. The summation of the cell-type percentages in all clusters is 100% for each method.

The identified pseudo-time-associated genes highlight the importance of the development and signaling of neurons across the cortical layers.

## Mouse cerebellum data by Slide-seq

Next, we analyzed the mouse cerebellum data[1], which contains 18,671 genes measured on 25,551 locations (Fig. 3). The RGB plot from SpatialPCA displays a clear regional segregation on the tissue, more clearly than the RGB plots from PCA or NMF (Supplementary Fig. 17a, b). The resulting RGB plots are not sensitive to the scaling of the input data (Supplementary Fig. 17c–f). In addition, genes associated with the spatial PCs are enriched in synapse-related, axon-related, synaptic signaling, and oligodendrocyte differentiation pathways (Supplementary Fig. 18 and Supplementary Table 6). Clustering based on the spatial PCs categorizes the cerebellum into eight distinct spatial domains that are consistent with the known anatomic structures of the cerebellum (Fig. 3b). These detected spatial domains include three sublayers of the granule cell layer (GCL), Purkinje cell layer, molecular layer, cerebellum nucleus, white matter, and choroid plexus (Fig. 3b and Supplementary Fig. 19). The clustering results from SpatialPCA are robust with respect to the number of spatial PCs used, the input gene set, the number of clusters, and the kernel matrix used for modeling spatial correlation (Supplementary Fig. 20). In addition, the spatial domains detected by SpatialPCA are spatially continuous and smooth (median LISI = 1.404, CHAOS = 0.016, PAS = 0.104), more so than that detected by the other methods (BayesSpace: median LISI = 3.489, CHAOS = 0.028, PAS = 0.781; SpaGCN: median LISI = 2.157, CHAOS = 0.021, PAS = 0.359; HMRF: median LISI = 2.326, CHAOS = 0.026, PAS = 0.479; PCA: median LISI = 3.166, CHAOS = 0.026, PAS = 0.528; NMF: median LISI = 3.877, CHAOS = 0.032, PAS = 0.845; Fig. 3c and Supplementary Fig. 17i, j).

A careful examination of the spatial domains detected by SpatialPCA and marker gene expression leads to two important observations. First, SpatialPCA identified an important spatial domain, choroid plexus, that is missed by the other methods (Fig. 3b and Supplementary Fig. 19). *Cst3*, a marker gene for choroid plexus[41], is clearly enriched in the choroid plexus identified by SpatialPCA (Fig. 3d). *Cst3* encodes the cystatin C protein, which is secreted primarily from the choroid plexus into the cerebrospinal fluid[42]. Second, unlike the other methods, the three sublayers of the GCL detected by SpatialPCA display expected spatial localization and transcriptomic profiles. Specifically, the outer GCL sublayer is adjacent to the Purkinje cell layer and is enriched with the marker gene *Otx2* (Fig. 3d). *Otx2* encodes orthodenticle homeobox 2, a transcription factor that shapes the morphogenesis of the cerebellum through controlling the proliferation of postnatal granule cell precursors[43]. The middle GCL sublayer is enriched with the marker gene *Cbln3*, which is predominantly expressed in mature granule cells that have ceased division and finished migration into the GCL[44]. CBLN3 forms a heteromeric complex with CBLN1 in the early postnatal stage and participates in CBLN-mediated synaptic development and function in the cerebellum[45–47]. The inner GCL sublayer is adjacent to the white matter and is enriched with the marker gene *Eomes*. *Eomes* encodes eomesodermin, also known as T-box brain protein 2 (TBR2), which is an important transcription factor expressed

in unipolar brush cells, a type of glutamatergic interneurons in GCL[48–50]. TBR2 is expressed in the nuclei of a subset of interneurons in the internal granular layer in adult mouse cerebellum[48] and these interneurons amplify inputs from vestibular ganglia and nuclei by spreading and prolonging excitation within the internal granular layer[48]. The correct identification of the choroid plexus and three GCL sublayers characterized by distinct transcriptomic profiles highlights the utility of SpatialPCA in revealing the transcriptomic and functional basis of fine-grained cerebellum structures.

We further examined the transcriptomic profile and cell-type compositions on the tissue regions detected by SpatialPCA. First, we performed DE analysis to identify regional-specific genes (Supplementary Table 10) and found them to be highly enriched in the pathways of neural nucleus development and synaptic signaling (Supplementary Fig. 21). The enriched pathways are critical for information processing and neural development in the cerebellar cortex[51,52]. Second, we performed deconvolution analysis to infer the cell-type composition on the detected spatial domains. As expected[53,54], we found that the detected Purkinje layer is enriched with Bergmann and Purkinje cells; the granule cell layer is enriched with granule cells; the white matter region is enriched with oligodendrocytes; and the cerebellum nucleus is enriched with interneurons (Fig. 3e and Supplementary Fig. 22). The Purkinje layer detected by SpaGCN, HMRF and BayesSpace is also enriched with Purkinje cells, though it is not as well segregated from the granule and molecular layers as in SpatialPCA (Fig. 3f, Supplementary Fig. 19, and Supplementary Fig. 22). In contrast, clusters inferred by NMF are dominated by granule cells while each PCA cluster is generally dominated by one cell type, supporting the utility of PCA in segregating cell types rather than spatial domains as demonstrated in simulations (Supplementary Fig. 22). We also examined the distribution pattern of representative cell types in the anatomic tissue structures. We reason that, if an anatomic tissue structure is correctly depicted, then its representative cell type should be enriched in this spatial domain. In all methods, the Bergmann cells are enriched in Purkinje layer as expected (Fig. 3g and Supplementary Figs. 22 and 23). However, in PCA, the Bergmann glia cells are wrongly clustered into the region of the molecular layer which is adjacent to the Purkinje layer. In SpaGCN and HMRF, the choroid plexus region is not detectable, such that the choroid plexus cells are distributed across multiple regions (Fig. 3g and Supplementary Figs. 22 and 23). While BayesSpace and NMF were not able to detect obvious Purkinje layer with clear boundaries.

## Mouse hippocampus data by Slide-seq V2

Next, we analyzed the mouse hippocampus data[2], which contains 23,264 genes measured on 53,208 locations (Fig. 4). Consistent with the previous datasets, the RGB plot from SpatialPCA displays a clear regional segregation of the tissue, more clearly than the RGB plots from PCA or NMF (Supplementary Fig. 24a, b). The resulting RGB plots are not sensitive to the scaling of the input data (Supplementary Fig. 24c–f). In addition, the genes associated with spatial PCs are enriched in synapse-related, cilium movement, electron transport, and neurogenesis

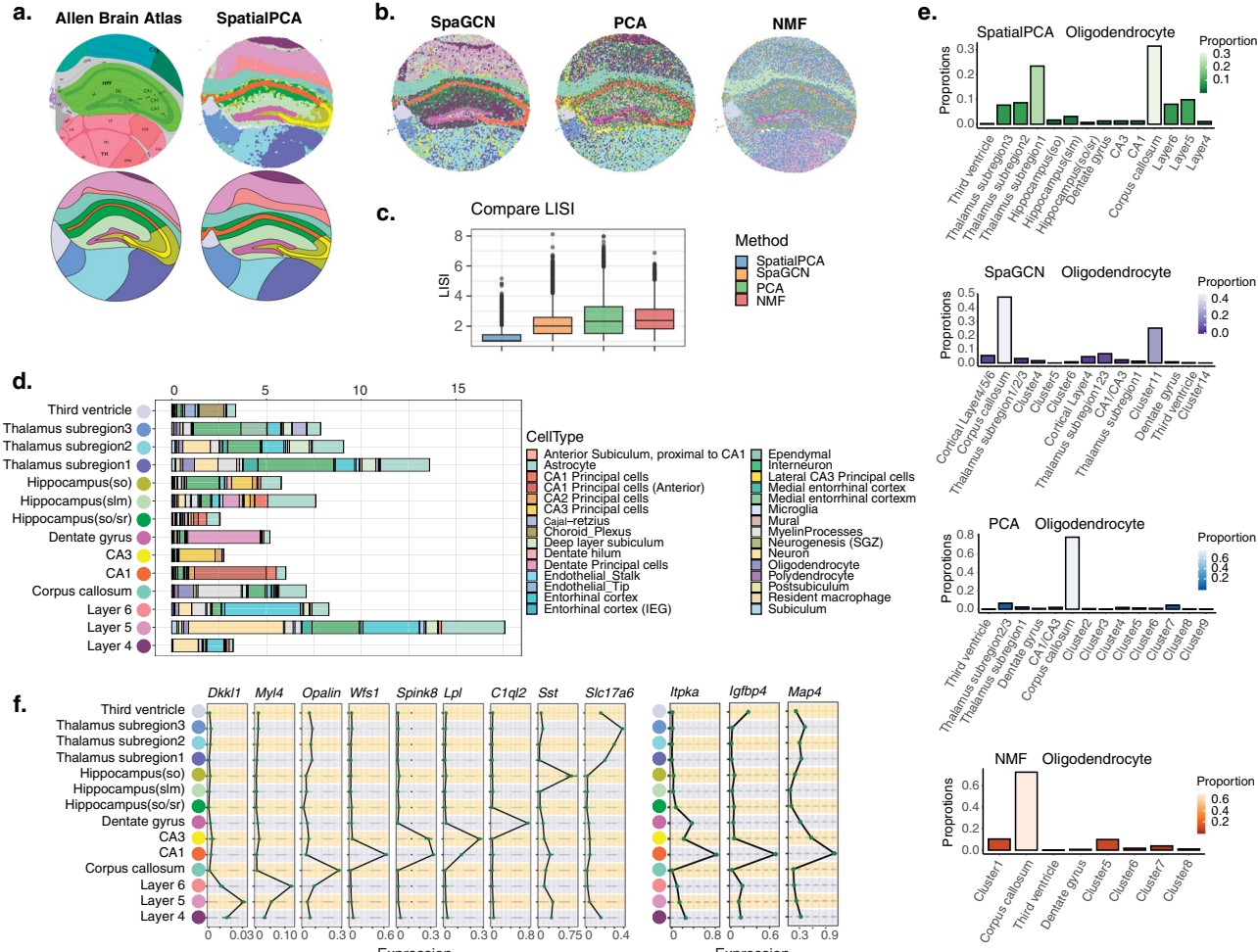

**Fig. 4 | Analysis of the hippocampus data from Slide-seq V2. a** Left panel: The structure of the mouse hippocampus, with the main tissue regions annotated from Allen Brain Atlas. Right panel: SpatialPCA segregates tissue locations into distinct tissue regions. Lower panel: Tissue structures based on annotations from Allen Brain Atlas and spatial domains detected by SpatialPCA. The detected tissue regions were annotated based on their relative positions on the tissue and the enriched cell types in each detected tissue domain. **b** Clustering of tissue locations based on the other methods. For SpaGCN, clusters were inferred directly by the software. For dimension reduction methods (PCA and NMF), clustering was performed based on the inferred low-dimensional components on spatially variable genes. **c** Clustering performance of different methods in obtaining smooth and continuous spatial domains measured by local inverse Simpson's index (LISI) in 51,398 locations. Lower LISI score indicates more homogeneous neighborhood spatial domain clusters of a spot. In the boxplot, the center line, box limits and whiskers denote the median, upper and lower quartiles, and 1.5× interquartile range, respectively. **d** Percentage of different cell types (x axis) in each spatial domain detected by SpatialPCA (y axis). **e** Distribution of oligodendrocyte cells in each cluster for different methods. The summation of the cell-type percentages in all clusters is 100% for each method. **f** Left: mean expression of marker genes in hippocampus (n = 51,398 locations). Right: mean expression of three differentially expressed genes in CA1 region that only detected by SpatialPCA clusters (n = 51,398 locations). In the boxplot, the center line denotes the mean value of the expression.

pathways (Supplementary Fig. 25 and Supplementary Table 6). Clustering based on the spatial PCs categorizes the hippocampus into fourteen distinct spatial domains that are consistent with the known anatomic structures of the hippocampus (Fig. 4a and Supplementary Fig. 26). These detected spatial domains include cortical layers (layers 4, 5, and 6), corpus callosum, CA1, CA3, dentate gyrus, hippocampus regions (so/sr, slm, so), thalamus regions (subregion 1, 2, and 3), and the third ventricle. The clustering results from SpatialPCA are robust with respect to the number of spatial PCs used, the input gene set, and the number of clusters (Supplementary Fig. 27). In contrast, clusters from SpaGCN, or clustering based on the low-dimensional components from PCA and NMF do not reveal clear segregation of spatial domains (Fig. 4b and Supplementary Fig. 26). Importantly, SpatialPCA also correctly detected the expected three cortical layers and three thalamus subregions while the other methods did not. In addition, the spatial domains detected by SpatialPCA are continuous and smooth (median LISI = 1.043, CHAOS = 0.013, PAS = 0.036), more so than those detected by the other methods (SpaGCN: median LISI = 2.014, CHAOS = 0.016, PAS = 0.031; PCA: median

LISI = 2.322, CHAOS = 0.020, PAS = 0.449; NMF: median LISI = 3.539, CHAOS = 0.021, PAS = 0.684; Fig. 4c and Supplementary Fig. 24i, j).

We carefully examined the transcriptomic profile and cell-type compositions in the spatial domains detected by SpatialPCA. First, we performed DE analysis to identify regional-specific genes (Supplementary Table 11) and found them to be highly enriched in the pathways of cellular components of the synapse as well as synapse signaling (Supplementary Fig. 28). Second, we performed deconvolution analysis to infer the cell-type composition on the detected spatial domains (Fig. 4d). As expected, we found that the detected CA1 region is enriched with CA1 principal cells; the CA3 region is enriched with CA3 principal cells; the dentate gyrus region is enriched with dentate principal cells; the third ventricle region is enriched with choroid plexus cells; and the cerebellum nucleus is enriched with interneurons. SpaGCN, PCA, and NMF were not able to distinguish between CA1 and CA3, as both regions are enriched with CA1 and CA3 principal cells (Supplementary Fig. 29). We then examined the distribution pattern of representative cell types in the anatomic tissue structures (Fig. 4e and

Supplementary Fig. 30). In SpatialPCA, the entorhinal cortex cells are enriched in the three detected cortical layers as expected. However, SpaGCN and PCA are not able to delineate the cortical layers, and the entorhinal cortex cells are enriched in clusters without continuous shape or boundaries. We did not compare representative cell-type distributions in NMF because it cannot detect obvious spatial domains with clear boundaries. Third, the spatial domains detected by SpatialPCA allowed us to identify region-specific markers that are not identified by other methods. For example, only SpatialPCA detected the enrichment of *Itpka*, *Igfbp4*, and *Map4* in the CA1 region of the hippocampus, consistent with their known enrichment pattern in the hippocampus pyramidal neurons[55–57] (Fig. 4f). Other marker genes that are highly expressed in the substructures of the hippocampus are shown in Fig. 4f.

We performed trajectory inference on the cortical layers 4–6 using spatial PCs and identified one trajectory projecting from layer 6 toward 4 (Supplementary Fig. 31a, b), which is again consistent with corticogenesis[39,40]. In contrast, the pseudo-time values inferred based on the low-dimensional components of PCA and NMF are intermingled across layers (Supplementary Fig. 31c, d). We further examined genes that are associated with the inferred pseudo-time by SpatialPCA. We found that the pseudo-time-associated genes are highly enriched in neuronal pathways such as synapse pathways and neurodegeneration diseases (Supplementary Fig. 16b and Supplementary Table 9), highlighting the functional importance of the trajectory inferred by SpatialPCA.

### HER2 tumor data by spatial transcriptomics (ST)

Finally, we analyzed the HER2-positive breast tumor data[58], which contains 15,030 genes measured on 613 spatial locations (Fig. 5). Consistent with previous datasets, the RGB plot by SpatialPCA clearly segregates the tissue into multiple spatial domains, with the neighboring locations sharing more similar colors than those that are far away (Supplementary Fig. 32a, b). The resulting RGB plots are not sensitive to the scaling of the input data (Supplementary Fig. 32c–f). In addition, genes associated with spatial PCs are enriched in cell activation, cell-cell adhesion, cell migration, and immune response pathways (Supplementary Fig. 33 and Supplementary Table 6). The top spatial PCs altogether are highly predictive of the tissue regional annotations based on the histological image obtained by pathologists in the original study (pseudo $R^2 = 0.666$), much more so than the low-dimensional components extracted from PCA (pseudo $R^2 = 0.477$), NMF (pseudo $R^2 = 0.472$; Fig. 5b). While the correlation between spatial PCs and regular PCs are high in this data (the absolute Pearson's correlation values between the top five spatial PCs and the top regular PCs are 0.953, 0.922, 0.933, 0.9, and 0.82), the spatial PCs still contain more spatial information than the regular PCs. The Moran's I of the top five spatial PCs are on average 30.2% higher than that of the corresponding regular PCs (range = 21–70%). Indeed, clustering based on the spatial PCs categorizes the tissue into seven spatial domains that are consistent with the regional annotations obtained by pathologists (ARI = 0.445, Fig. 5c, d). The seven detected spatial domains include normal glands, cancer region, cancer surrounding region, fibrous tissue near tumor, immune cell region, fat tissue, and fibrous tissue near normal gland (Fig. 5c). The spatial domains detected by SpatialPCA are more accurate than that by the other two dimension reduction methods (PCA: ARI = 0.342; NMF: ARI = 0.32) and that by the three spatial domain detection methods (BayesSpace: ARI = 0.418; SpaGCN: ARI = 0.376; HMRF: ARI = 0.35, Fig. 5d). The clustering results from SpatialPCA are robust with respect to the number of spatial PCs used, the input gene set, and the kernel matrix used for modeling spatial correlation (Supplementary Fig. 34). In addition, the spatial domains detected by both SpatialPCA (median LISI = 1.84, CHAOS = 0.138, PAS = 0.198) and BayesSpace (median LISI = 1.73, CHAOS = 0.139, PAS = 0.176) are continuous and smooth, more so than those detected

by the other methods (SpaGCN: median LISI = 2, CHAOS = 0.153, PAS = 0.265; HMRF: median LISI = 1.95, CHAOS = 0.149, PAS = 0.272; PCA: median LISI = 1.94, CHAOS = 0.148, PAS = 0.354; NMF: median LISI = 2.25, CHAOS = 0.154, PAS = 0.437; Supplementary Fig. 35). The comparison results on spatial domain detection not only hold for the above sample H1 but also for all eight tissue sections where the original study provided ground truth spatial domain annotations (Supplementary Fig. 36). We further evaluated the ability of different methods in detecting fine-grained structures on the tissue by examining the expression pattern of domain-specific genes. In particular, we calculated an enrichment score of the domain-specific genes in the domains detected by different methods, where a high score suggesting a consistent pattern between the domains detected by the method and those characterized by the domain-specific genes (details in the Methods; Supplementary Fig. 37). As expected, SpatialPCA achieves the highest enrichment score (1.417) compared with the other methods (BayesSpace 1.269; HMRF 1.202; NMF 1.345; PCA 1.248; SpaGCN 1.096) not only in the sample H1 but also for all eight annotated samples (SpatialPCA median score 1.141; BayesSpace 1.082; HMRF 0.916; NMF 0.983; PCA 0; SpaGCN 0.822; Supplementary Fig. 36b–d). Similar to the application in DLPFC, the histological characteristic vector in the ST tumor data is not that predictive of the pathologist annotations (pseudo $R^2 = 0.258$). Consequently, when histological information was included in SpatialPCA, we observed decreased accuracy for spatial domain detection (ARI = 0.254, Supplementary Fig. 14c, d).

We characterized the transcriptomic and cell compositional properties of the spatial domains detected by SpatialPCA through two additional analyses. First, we identified genes that are specifically expressed in different spatial domains through DE analysis (Supplementary Table 12). We found that the DE genes in the immune region detected by SpatialPCA are highly enriched in the pathways of immune response, while the DE genes in the tumor region detected by SpatialPCA are highly enriched in the pathways of biological adhesion (Supplementary Fig. 38). Second, we carefully examined the cell-type compositions in each spatial domain through cell-type deconvolution (Fig. 5e and Supplementary Fig. 39). In the analysis, we found that the detected tumor region mainly contains epithelial basal cells and epithelial basal cycling cells, which are cancer cells and cancer cells with high proliferation, respectively[59]. The detected tumor surrounding region is mainly composed by cancer cells with high proliferation, along with some B cells and T cells. In PCA and NMF, the detected tumor regions are not as smooth or continuous as SpatialPCA, as the latter explicitly models spatial correlation in dimension reduction. In addition, PCA and NMF inferred the original annotated tumor region to consist of two or more clusters while not being able to detect the fat tissue region. Importantly, we found that the detected immune cell region resembles a tertiary lymphoid structure (TLS)[60,61] with multiple features of TLS: the region is located near the tumor; the region primarily contains T cells and B cells, all of which are key cell types in TLS[60,62–64]; the region is enriched with chemokine signature genes such as *CCL19* and *CCL21* and T follicular helper cell signature genes such as *CXCL13* and *TIGIT* (Fig. 5f), all of which are marker genes associated with TLS neogenesis in breast cancer[60]. We also obtained a TLS score on each location from the original study, which were computed based on the interaction strength between B cells and T cells as a key indicator of TLS. We found that the TLS scores are highly enriched in the TLS region detected by SpatialPCA (Supplementary Fig. 40). TLS is an ectopic lymphoid organ developed in non-lymphoid tissues that generate and regulate antitumor defenses. The detection of TLS is predictive of the treatment outcome in HER2-positive tumors[65] and highlights the potential of SpatialPCA in understanding antitumor immune response and future prediction of tumor outcome.

The accurate tissue domains detected by SpatialPCA allowed us to identify multiple region-specific genes. For example, we found

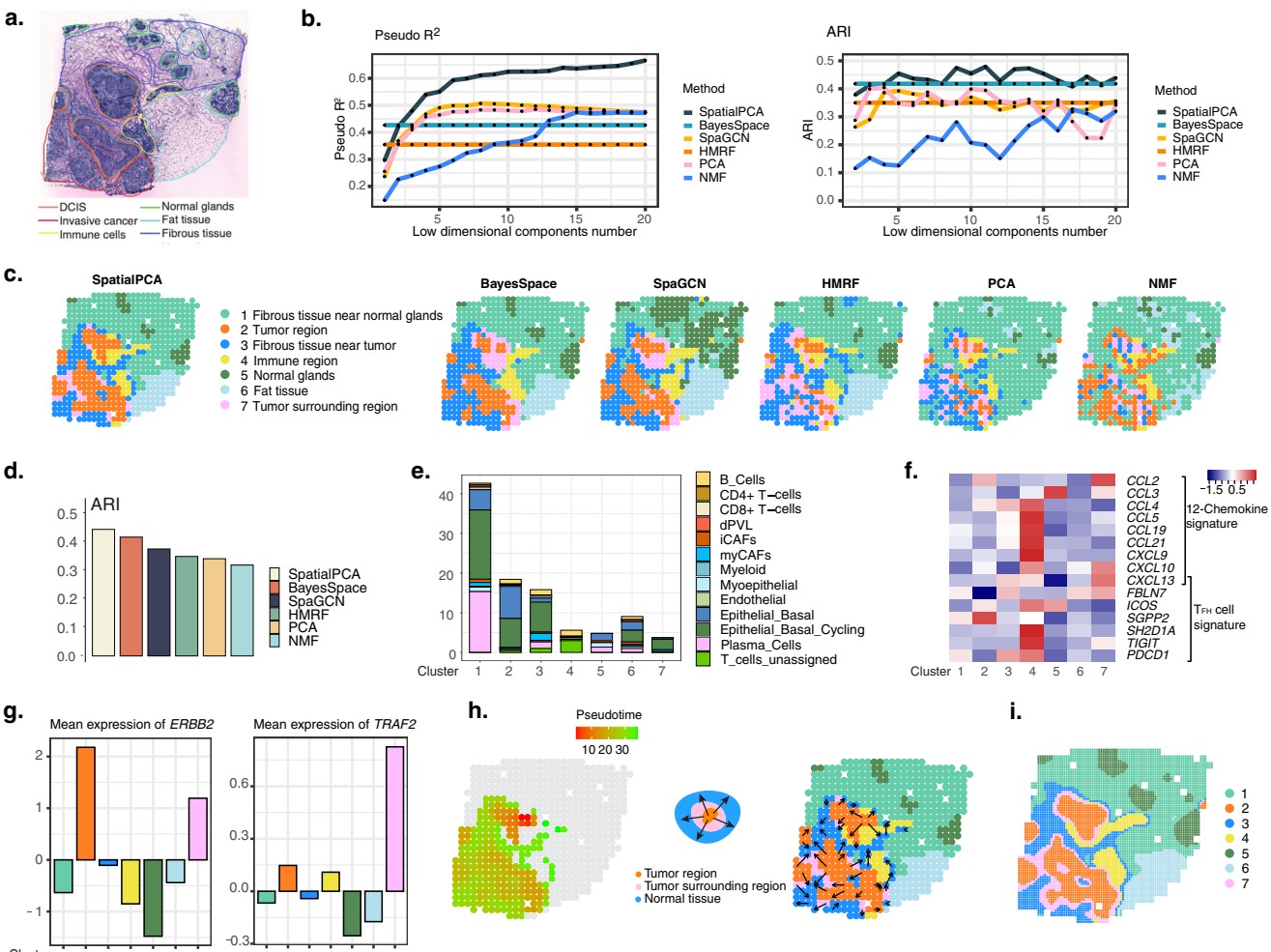

**Fig. 5 | Analysis of the HER2 tumor data from ST. a** Haematoxylin and eosin (H&E) staining image shows distinct tissue regions annotated by a pathologist in the original study. The annotated tissue regions include invasive cancer (red), fat tissue (cyan), fibrous tissue (blue), normal breast glands (green), in situ cancer/DCIS (orange), and immune cells (yellow). **b** Left: spatial PCs in SpatialPCA have higher prediction accuracy in terms of McFadden's pseudo-$R^2$ (right) for the ground truth labels than latent components in PCA, NMF, and SpaGCN. For BayesSpace and HMRF, we treated their inferred cluster labels as predictors. Right: spatial PCs in SpatialPCA have higher spatial domain clustering accuracy in terms of adjusted Rand index (ARI) for the ground truth labels. **c** Clustering of tissue locations using different methods. For dimension reduction methods (SpatialPCA, PCA, and NMF), clustering was performed based on the inferred low-dimensional components on spatially variable genes. For BayesSpace, SpaGCN, and HMRF, clusters were inferred directly by the software. **d** Clustering accuracy of different methods in

recapitulating the true tissue regions. Accuracy is measured by ARI. **e** The percentage of cell types inferred through cell-type deconvolution analysis on each tissue domain detected by SpatialPCA. **f** Heatmap shows the mean expression level of multiple tertiary lymphoid structure (TLS) signature genes on different tissue domains detected by SpatialPCA. **g** Mean expression of tumor region differentially expressed gene *ERBB2* and tumor surrounding region differentially expressed gene *TRAF2*. The figure legend on the bottom relates the cluster numbers detected by SpatialPCA to tissue domain names. **h** Left: visualization of the trajectory inferred by SpatialPCA. Middle: diagram shows the directionality of the trajectory, which points from the tumor region toward the normal tissue through the tumor's surrounding region. Right: arrows point from tissue locations with low pseudo-time to tissue locations with high pseudo-time. **i** High-resolution spatial map reconstruction in SpatialPCA.

that *ERBB2* is enriched in the tumor region detected by SpatialPCA while *TRAF2* is enriched in the tumor surrounding region detected by SpatialPCA (Fig. 5g). *ERBB2* encodes a membrane receptor in the epidermal growth factor receptor family, is a key breast cancer marker, and involves in breast cancer metastasis[66–72]. *TRAF2* is associated with breast cancer progression and metastasis[73], is a member of the tumor necrosis factor receptor-associated factor family of intracellular signal transduction proteins, and is a critical mediator of both the activation of NF-κB and MAPK pathways[74]. The enrichment of *TRAF2* in the tumor surrounding region suggests the progression of tumor cells in invading and penetrating the surrounding tissues, potentially aiding in the prognosis of the cancer stage.

We performed trajectory inference on tumor and tumor-adjacent regions to investigate how the tissue locations are connected to one

another in tumorigenesis. Spatial trajectory inference based on spatial PCs identified one trajectory pointing from tumor region towards tumor surrounding region and further towards normal tissues (Fig. 5h). We further examined genes that are associated with the inferred pseudo-time (Supplementary Fig. 41 and Supplementary Table 9). We found that the pseudo-time-associated genes are highly enriched in immune response, cell-mediated immunity, and phagocytosis recognition pathways, highlighting their importance in cancer progression, tumor invasiveness, and metastasis[75,76].

## Discussion

We have presented SpatialPCA, a spatially aware dimension reduction method that is tailored for spatial transcriptomics. SpatialPCA explicitly models the spatial correlation structure in the latent space during dimension reduction and preserves the neighborhood similarity in the

original data onto the low-dimensional manifold. As a result, the low-dimensional components of SpatialPCA contain valuable spatial correlation information and can be paired with existing scRNA-seq analysis tools to enable various downstream analysis of spatial transcriptomics. In addition, SpatialPCA relies on a data-generative model to accommodate spatial correlation and can thus be used to impute the low-dimensional components on new and unmeasured tissue locations, enabling the construction of a refined spatial map with increased spatial resolution. SpatialPCA is computationally efficient, and is easily scalable to spatial transcriptomics with tens of thousands of spatial locations and thousands of genes (Supplementary Table 13). We have illustrated the benefits of SpatialPCA for spatial transcriptomics visualization, spatial domain detection, trajectory inference on the tissue, as well as high-resolution spatial map construction.

While we have primarily focused on spatial domain detection and trajectory inference on the tissue, we note that the modeling framework of SpatialPCA can also allow us to impute low-dimensional components and gene expression levels on new tissue locations, thus facilitating the construction of a high-resolution spatial map on the tissue (details in "Methods"). Because of the data-generative nature of SpatialPCA and its explicit modeling of spatial correlation, SpatialPCA can be used to construct the high-resolution spatial map for any spatial transcriptomics technologies with any arbitrarily high resolution—both these features are in direct contrast to BayesSpace, which can only enhance ST or 10x Visium data with a fixed resolution of either six or nine times higher than that of the original. We applied SpatialPCA to construct a high-resolution map in the low-resolution ST tumor data and found that the constructed high-resolution map displays continuous and spatially smooth pattern, more so than that produced by BayesSpace (Fig. 5i and Supplementary Fig. 42). Clustering based on the high-resolution spatial map also pinpoints precise boundaries between tissue regions and refines a thin layer of tumor surrounding region immediately outside both the tumor and immune regions, which highlights the ability of SpatialPCA in detecting fine-grained transcriptomic changes that underlie the structural and spatial organization of carcinogenesis.

We have primarily focused on analyzing normalized data for dimension reduction. The raw gene expression measurements from spatial transcriptomics, however, are obtained in the form of counts:[77] they are collected either as the number of barcoded mRNA for a given transcript in each single-cell through smFISH-based technologies or as the number of sequencing reads mapped to a given gene on each tissue location through sequencing-based spatial technologies. Analyzing normalized expression data from spatial transcriptomics can be suboptimal as this approach fails to account for the mean–variance relationship existed in raw counts, leading to a potential loss of inference accuracy and subsequent loss of analysis power. Indeed, a similar loss of power has been well documented for methods that only analyze normalized data in many other omics sequencing studies[78–81] as well as in spatial transcriptomics[77]. Consequently, many recently developed dimension reduction methods for scRNA-seq studies have chosen to directly model raw count data, which have resulted in improvement in inference accuracy[20–22]. In principle, SpatialPCA can be extended to model raw count data from spatial transcriptomics based on the generalized linear model framework, with an additional zero component to model the potential zero inflation that might be encountered in certain spatial transcriptomic technologies. Such extension of SpatialPCA, however, will likely incur substantial increase in computational cost, along with potentially numerical instability issues associated with optimizing the likelihood function in the presence of sparse counts[82]. Extending SpatialPCA towards effective modeling of count data while keeping computational cost and numerical stability in check will be an important future research direction.

We have primarily performed SpatialPCA analysis using kernels computed based on Euclidean distance. We note, however, that the framework of SpatialPCA is flexible and can be paired with kernels computed using various non-Euclidean distances. To illustrate this feature, we performed analysis in the DLPFC dataset using Delaunay triangulation-based distance (details in "Methods"). We found that SpatialPCA with the Delaunday distance achieves an ARI of 0.453 for spatial domain detection, which is lower than using the default covariance matrix constructed by Euclidean distance (ARI = 0.577, Supplementary Fig. 10b). While non-Euclidean distance does not work as well as Euclidean distance in this particular dataset, we note that non-Euclidean distance could be beneficial in other datasets and thus we include Delaunday distance as an option in the SpatialPCA software. For trajectory inference, we have followed recent approaches[83,84] of directly applying single-cell RNA-seq trajectory inference methods to spatial transcriptomics datasets. We note, however, that the single-cell trajectory inference methods were initially designed with single-cell data in mind, and it may not be optimal to directly apply them to spatial transcriptomics where the measured locations can contain a mixture of cells[85]. Therefore, it will be important in the future to develop spatial transcriptomics-specific trajectory inference methods, which, when paired with spatial PCs, may further improve the performance of trajectory inference. SpatialPCA can also be easily extended to perform dimension reduction on multiple samples (details in "Methods"). We applied such extension of SpatialPCA to jointly analyze three tissue sections in the DLPFC data, with one from each of the three individuals. In the joint analysis, we found the extension of SpatialPCA can help substantially improve the spatial domain detection accuracy for some tissue sections but at the slight cost of the accuracy loss of other sections. Specifically, the single dataset analysis version of SpatialPCA achieves an ARI of 0.540, 0.376, and 0.577 for samples 151507, 151669, and 151673, respectively; while the extension of SpatialPCA achieves an ARI of 0.518, 0.431, and 0.552 (Supplementary Fig. 10c). Finally, we note that SpatialPCA can also be applied to datasets collected from in situ hybridization-based technologies. To illustrate such feature, we applied all methods to a MERFISH dataset[86] (Supplementary Fig. 43) and found that SpatialPCA achieved higher spatial clustering accuracy (median ARI = 0.454) than other methods (median ARI: BayesSpace 0.1, SpaGCN 0.262, HMRF 0.414, NMF 0.06, PCA 0.07).

We have primarily focused on performing spatially aware dimension reduction based on the probabilistic version of PCA. PCA is a linear dimension reduction method that effectively expresses the low-dimensional components as a linear function of the input matrix. Despite its simplicity, dimension reduction based on PCA is surprisingly effective and facilitates many downstream analytic tasks in scRNA-seq studies[16]. Indeed, PCA is implemented in commonly applied scRNA-seq tools such as Seurat[23], SCANPY[87], and Cell Ranger R Kit[88], for downstream data visualization, clustering analysis, or trajectory inference. Certainly, dimension reduction based on linearity likely only captures the first-order relationship between the original data and the reduced manifold and may not be effective in capturing all complex biological signals contained in the input genomic data. To further improve the effectiveness of dimension reduction and capture non-linear signals in the expression matrix, several non-linear dimension reduction methods have been recently developed for scRNA-seq. These non-linear methods are often based on deep neural networks and can flexibly model the non-linear relationship between the expression matrix and the extracted low-dimensional components[89–92]. Extending SpatialPCA towards non-linear modeling based on the deep learning framework for spatial transcriptomics is another important future direction.

Finally, we note that the main idea of incorporating spatial correlation information into PCA has a long-standing history in geographics and genetics. In geographics, PCA has been commonly applied for dimension reduction in geographical datasets[93] and has

been previously extended to a geographically weighted version that performs local PCA at each spatial location while accommodating the local neighboring structure[94]. In genetics, a spatial version of PCA was previously developed to incorporate the autocorrelation structure measured by Moran's I into PCA for the identification of cryptic spatial patterns of genetic variability[95]. Our SpatialPCA effectively extends these ideas into a formal data-generative model with a maximum likelihood inference framework. Compared to these early approaches, SpatialPCA performs dimension reduction at the global instead of local level, automatically infers the contribution of spatial correlation in determining the variation in the low-dimensional components, and is capable of imputing low-dimensional components in new and unmeasured spatial locations. Because of these modeling advantages, examining the utility of SpatialPCA in applications beyond spatial transcriptomics may benefit other research fields.

## Methods

### SpatialPCA overview

We consider a spatial transcriptomics study that collects gene expression measurements for $m$ genes on $n$ spatial locations of a tissue. These locations have known spatial coordinates that are recorded during the experiment. We denote $s_i$ as the $k$-vector of spatial coordinates for $i$th location, with $i \in (1, \ldots, n)$. Depending on the spatial transcriptomics technology, the spatial coordinates vary continuously over either a two-dimensional space ($k = 2$; $s_i = (s_{i1}, s_{i2}) \in R^2$) or a three-dimensional space ($k = 3$; $s_i = (s_{i1}, s_{i2}, s_{i3}) \in R^3$). We denote $Y$ as the $m \times n$ gene expression matrix measured in the study. The $ji$th element of $Y$, $y_{ji}(s_i)$, represents the gene expression measurement for $j$th gene on $i$th location. Following[88,96,97], we assume that the expression measurements have already been normalized through variance stabilizing transformation and further scaled for each gene to have zero mean and unit standard deviation.

Our goal is to perform dimension reduction on the gene expression matrix and infer a $d \times n$ factor matrix $Z$ that represents a low-dimensional embedding of $Y$. The factor matrix $Z$ contains $d$ factors, and its $l$th row, $Z_l$, is an $n$-vector that represents the $l$th factor values across $n$ locations. For dimension reduction, we consider the following latent factor model

$$Y = (XB)^T + WZ + E \tag{1}$$

where $X$ is an $n \times q$ matrix of covariates, which contains a column of ones as the intercept along with ($q - 1$) other potential covariates; $B$ is a $q \times m$ matrix of corresponding coefficients; $W$ is a $m \times d$ factor loading matrix; and $E$ is an $m \times n$ matrix of residual errors. In the present study, we only include the intercept for all analyses, and thus $q = 1$. We assume that the $ji$th element of $E$, $E_{ji}$, follows an independent normal distribution with mean zero and variance $\sigma_0^2$, or $E_{ji} \sim N(0, \sigma_0^2)$.

The factor model in Eq. (1) is not yet identifiable as any rotational transformation of $W$ and $Z$ would lead to the same solution. Therefore, we need to place further modeling constraints on both $W$ and $Z$ to ensure model identifiability. For $W$, we follow the probabilistic principal component analysis model (PPCA)[98] and impose an orthonormality constraint on its columns to have $W^T W = I_d$. For $Z$, PPCA[98] and other previous approaches[99,100] commonly assume element independence. That is, each element of $Z$ is independently and identically distributed from a normal distribution $N(0,1)$. However, such element independence assumption on $Z$ is not ideal for spatial transcriptomics. In spatial transcriptomics, the neighboring locations on a tissue often share similar composition of cell types and display similar gene expression levels. Consequently, the factor values on neighboring locations are likely similar to each other, more so than those on locations that are far away. The factor values on neighboring locations thus contain invaluable information that can be used to facilitate the inference of the factor values on the location of interest. To encourage

neighborhood similarity in factor values and facilitate information sharing across neighboring locations for factor estimation, we follow[101] and assume that each $Z_l$ follows a multivariate normal distribution

$$Z_l. \sim MVN(0, \Sigma_l) \tag{2}$$

where the $n \times n$ covariance matrix $\Sigma_l$ models the correlation among the spatial locations and is a function of their spatial coordinates. Here, we use the Gaussian kernel to construct the covariance matrix and assume that the $(i,i')$th entry of $\Sigma_l$ is in the form of $\sigma_0^2 \tau_l K(s_i, s_{i'})$, where $K(s_i, s_{i'}) = \exp(-|s_i - s_{i'}|^2 / \gamma)$ with $\gamma$ being the bandwidth parameter and $\sigma_0^2 \tau_l$ being a variance component that is scaled with respect to the residual error variance $\sigma_0^2$. The functional form of $\Sigma_l$ is designed to induce spatial correlation of factor values on the tissue and encourage factor similarity in neighboring locations. Specifically, if two locations are close to each other, then the corresponding element in $\Sigma_l$ will be large, leading to similar factor values on the two locations; and vice versa. The bandwidth parameter $\gamma$ in $\Sigma_l$ determines the strength of such spatial correlation: a small $\gamma$ leads to a low spatial correlation while a large $\gamma$ leads to a high spatial correlation[102,103]. The variance parameter $\tau_l$ in $\Sigma_l$, on the other hand, determines the scale of the $l$th factor values: a small $\tau_l$ corresponds to small factor values relative to the residual errors while a large $\tau_l$ corresponds to large factor values relative to the residual errors. For $\gamma$, we determine $\gamma$ in a sample size dependent fashion[104,105]. Specifically, for data with large samples ($n > 5000$), we use Silverman's "rule-of-thumb" bandwidth, which is defined as $0.9 \min(\hat{\sigma}, \frac{IQR}{1.34}) n^{-\frac{1}{5}}$. Here, the bandwidth is computed for one gene at a time, where $\hat{\sigma}$ is the standard deviation of gene expression and IQR is the interquartile range of gene expression[104]. We obtain the median bandwidth value across all genes to serve as $\gamma$. For a data with small samples, the asymptomatic "rule-of-thumb" bandwidth is no longer applicable. Thus, we use the non-parametric Sheather & Jones's bandwidth that is especially robust for data with small samples[105]. We again compute the bandwidth for each gene in turn and obtain the median value across all genes to serve as $\gamma$. For $\tau_l$, we follow PPCA[98] and assume scale homogeneity across factors by setting $\tau_l = \tau$.

With the above model specifications, we infer the factor loading matrix $W$ and the factor matrix $Z$, along with the hyper-parameters ($\tau, \sigma_0^2$), through maximum likelihood-based optimization. Specifically, we first integrate out both $B$ and $Z$ to obtain a marginal likelihood, based on which we infer $\tau, \sigma_0^2$ and $W$ (details in the Supplementary Note). We then estimate $Z$ by computing their posterior mean conditional on the estimated $\tau, \sigma_0^2$ and $W$. In the algorithm, we incorporate multiple algebraic innovations to enable scalable computation such as applying a low-rank approximation on the kernel matrix $K$, with a low rank $r$ to ensure that the approximate matrix captures at least 90% of the variance in the original matrix in the present study. Overall, the computational time complexity of our algorithm is $O(tdm^2 + rn^2)$, where $t$ represents the number of iterations used in the optimization algorithm, with memory requirement being $O(mn + n^2)$ (Details in Supplementary Note). To further save memory and computation time, we also provide in the software the option to calculate a sparse kernel matrix when the sample size is large. The user can determine the sparsity level of the matrix by providing a cutoff value (default 1e-20) to set every element below this value to be 0. Because the factor model builds upon PPCA and the factor matrix $Z$ contains crucial spatial correlation information among locations, we refer to our model as the spatial probabilistic PCA (SpatialPCA) and refer to these factors as spatial PCs. SpatialPCA is implemented in an R package, which uses an S4 object to contain the model parameters and takes raw expression count data and spatial locations as input. SpatialPCA is freely available at www.xzlab.org/software.html with detailed documentation.

## Downstream analyses with spatial PCs

The inferred spatial PCs $Z$ from SpatialPCA can be paired with various methods already developed in the scRNA-seq literature to enable a range of downstream applications in spatial transcriptomics. Following ref. 33, we examine the performance of SpatialPCA through downstream analyses. Here, we examine the use of SpatialPCA for two important analytic tasks in spatial transcriptomics: spatial domain detection and spatial trajectory inference on the tissue. Spatial domain detection aims to segment the tissue into multiple structures, domains or micro-environments, each of which is characterized by a distinct transcriptomic profile. Spatial trajectory inference, on the other hand, aims to infer the transcriptomic relationship among spatial locations and construct trajectories directly on the tissue to represent the potential developmental lineages across locations that happened in the past. To the best of our knowledge, only a handful methods have been developed for spatial domain detection[28,29,31] and no method has been published for spatial trajectory inference on the tissue. For spatial domain detection, we formulate it as a clustering problem on the inferred spatial PCs $Z$. In particular, we apply standard scRNA-seq clustering algorithms on $Z$ to categorize spatial locations into different spatial domains. Because $Z$ contains critical spatial correlation information across locations, clustering based on the spatial PCs would lead to similar cluster assignments in the neighboring locations, resulting in smooth boundaries in the detected tissue structures. For spatial trajectory inference on the tissue, we apply standard scRNA-seq trajectory analysis methods on $Z$ to infer the progression in gene expression across spatial locations. Spatial trajectory inference on $Z$ allows us to properly account for the spatial relationship among locations, resulting in continuous trajectories with consistent directionality across neighboring locations. In the present study, we use Walktrap algorithm and Louvain algorithm for clustering analysis and slingshot[19] for trajectory analysis following the recommendations of[18,35]. However, SpatialPCA can be potentially paired with any clustering algorithm[16] or trajectory inference algorithm[18] developed in the scRNA-seq literature to take advantage of their benefits.

Importantly, because SpatialPCA builds upon a data-generative model that automatically infers the spatial correlation structure across tissue locations, it can also be used to construct a refined spatial map on the tissue. In particular, with the inferred spatial correlation, we can predict and impute spatial PCs on new and unmeasured locations based on the inferred spatial PCs on the measured locations. Imputing spatial PCs on new locations would allow us to obtain a spatial map with a resolution much higher than that measured in the original study. To do so, we examine one spatial PC at a time. For the $l$th spatial PC, we denote $Z_l(s)$ as the $n$-vector of spatial PC values on the original locations $s$ and denote $\widetilde{Z}_l(\widetilde{s})$ as the $\widetilde{n}$-vector of spatial PC values on the $\widetilde{n}$ new locations $\widetilde{s}$. Based on Eq. (2), the $(n + \widetilde{n})$-vector of $(Z_l(s), \widetilde{Z}_l(\widetilde{s}))^T$ follows a multivariate normal distribution. Consequently, we can obtain the conditional mean of $\widetilde{Z}_l(\widetilde{s})$ given $Z_l(s)$ based on the property of multivariate normal distribution:

$$\widetilde{Z}_l(\widetilde{s}) = \widetilde{\Sigma}_l^T(\widetilde{s}, s)\Sigma_l^{-1}Z_l(s) \tag{3}$$

where $\widetilde{\Sigma}_l(\widetilde{s}, s) = \sigma_0^2 \tau K(\widetilde{s}, \widetilde{s})$ is an $\widetilde{n}$ by n covariance matrix measuring the spatial correlation between the new and measured locations. We use the conditional mean in Eq. (3) to serve as the imputed spatial PCs on the new locations.

The imputed PCs on the new locations also allow us to directly impute the gene expression levels for individual genes on the new locations. Specifically, we calculate the posterior mean of gene expression as the product of the estimated loading matrix $\widetilde{W}$ and the high-resolution predicted spatial PCs $\widetilde{Z}_l(\widetilde{s})$, with

$$\widetilde{Y} = \widetilde{W}\widetilde{Z}_l(\widetilde{s}) \tag{4}$$

## Simulations

We performed simulations to evaluate the performance of our method and compare it with the other methods. To do so, we first obtained the cortex tissue from the DLPFC data (sample id 151673) and manually segmented the tissue into four cortical layers through Adobe Illustrator as illustrated in Fig. 1a. We exported the illustrative figure as a 1100 pixel by 984 pixel image in JEPG format and extracted the four cortical layer labels for each pixel based on the RGB values. We then randomly sampled 10,000 pixels from the image to serve as the single-cell locations and extracted their x/y coordinates. We assumed that these single cells belong to four different cell types, with distinct cell-type composition in each cortical layer. In particular, we set the cell types with the highest to lowest proportions to be 1, 2, 3, and 4 in the first layer, 2, 3, 4, 1 in the second layer, 3, 4, 1, 2 in the third layer, and 4, 1, 2, 3 in the fourth layer. We then considered four cell-type composition scenarios in the simulations. In the first scenario, each layer contains one dominant cell type, with 70% of the cells belonging to the dominant cell type and 10% of the cells belong to each of the three minor cell types. In the second scenario, each layer contains two dominant cell types with equal proportion, each consisting of 45% of cells, along with two minor cell types each consisting of 5% of cells. In the third scenario, each layer contains two major cell types with unequal proportion, with one consisting of 60% of cells and the other consisting of 30% of cells, along with two minor cell types each consisting of 5% of cells. In the fourth scenario, each layer contains three major cell types and one minor cell type, consisting of 35%, 30%, 30%, or 5% of cells, respectively. In each scenario, we randomly assigned each single cell in each layer to one of the four cell types based on a multinomial distribution with parameters set to be the cell-type composition in the layer. We performed ten simulation replicates for each simulation scenario.

In parallel, we obtained a scRNA-seq data on human prefrontal cortex obtained through Smartseq2[106], which contains expression measurements for 24,153 genes and 2394 cells that belong to six cell types annotated in the original study. The six cell types include astrocytes ($n = 76$ cells), GABAergic neurons ($n = 701$), microglia ($n = 68$), neuron cells ($n = 1057$), oligodendrocyte progenitor cell (OPC; $n = 117$), and stem cells ($n = 290$). We used the cells from the neuron cells in the scRNA-seq data as a reference to simulate gene expression counts for 5000 genes and 30,000 single cells using Splatter[107]. In Splatter, we used the splatEstimate function to estimate the cell-type parameters in the scRNA-seq data; we set de.prob=0.5; and we set group.prob=c(0.25, 0.25, 0.25, 0.25) so that the four cell types have equal probability. From the simulated cells, we randomly selected 10,000 of them with the desired cell-type compositions determined by the simulation scenario described in the above paragraph and assigned these cells onto the 10,000 locations to create the single-cell resolution spatial transcriptomics. With the single-cell resolution spatial transcriptomics, we further generated spot-level spatial transcriptomics data by merging expression counts of single cells into spots. In particular, we created square grids on the tissue, treated each square grid as a subspot, and annotated the spatial domain label of each subspot based on the majority of the spatial domain labels for cells located within the subspot. We treated the spatial domain label for the subspots as ground truth for spatial clustering accuracy comparison. We merged every nine subspots into a spot following the 10X ST subspot layout and obtained the coordinates for the spot based on the coordinates of the center subspot[31]. We varied the length of the square grids to create three spot-level spatial transcriptomics settings with varying resolution, with spot diameter being 90 μm ($n = 5077$ spots), 107 μm ($n = 3602$), or 145 μm ($n = 1948$); where the spot diameter for a 10X ST data is 100 μm. We focused on the simulation setting with spot diameter being 90 μm in the main text and places the other two cases in Supplementary Figures.

For each simulated spatial transcriptomics, we applied SpatialPCA and the other methods to detect spatial domains on the tissue. The other methods include BayesSpace, SpaGCN, HMRF, PCA, and NMF. We did not compare with stLearn in the simulations as the simulations did not contain H&E images−stLearn becomes PCA without an H&E image. Following the recommendation of SPARK-X[82], we obtained spatially variable genes (SVGs) in the single-cell level simulations using SPARK-X due to its efficiency in computational speed and memory cost and obtained SVGs in the spot-level simulations using SPARK due to its higher statistical power.

Besides the above main simulation scenarios, we also performed additional simulations to examine the performance of SpatialPCA and the other methods in detecting spatially non-contiguous and non-smoothly varying domains such as stripe patterns (Supplementary Fig. 4f). In particular, we divided the tissue into six equal-sized stripes and denoted the three odd stripes as domain one and the three even stripes as domain two. We set the proportions of the four cell types to be different in the two domains: in domain one, the cell types with the highest to lowest proportions are cell types 1, 2, 3, and 4; in domain two, the four types with the highest to lowest proportions are cell types 2, 3, 4, 1. We considered three cell-type composition scenarios similar to those used in the above main simulations. Specifically, in the first scenario, each layer contains two dominant cell type, with 60% of the cells belonging to the dominant cell type and 40% of the cells belong to the other cell type. In the second scenario, each spatial domain contains two major cell types with unequal proportion, with one consisting of 60% of cells and the other consisting of 30% of cells, along with two minor cell types each consisting of 5% of cells. In the third scenario, each spatial domain contains three major cell types, with one consisting of 60% of cells, along with two each consisting of 20% of cells. As above, we performed ten simulation replicates for each simulation scenario.

Besides the above main simulation strategy, we also performed simulations using a cell split strategy, where we introduced additional spatial correlation between spots to simulate the sharing of counts among neighboring spots in non-single-cell level spatial transcriptomics. Specifically, we first obtained 10,000 cells and assigned them to the spatial locations in the same four scenarios described in the main simulations. In the process, we randomly selected 2500 cells to be split. For each of these cells, we randomly split its expression level into four components based on uniform random weights that sum up to one. We then added each of the four components randomly to the expression level of one of its four nearest neighboring cells. This way, we obtain a total of 7500 pseudo-cells, 86.5% across simulation replicates contain split expression level from some of its neighboring cells. Afterward, we aggregate the 7500 pseudo-cells into spots. Again, we performed ten simulation replicates for each simulation scenario using this cell split simulation strategy.

### Analyzed datasets
We examined four public spatial transcriptomics datasets that include the followings.

**DLPFC human prefrontal cortex data by Visium.** We downloaded 12 human DLPFC[35] tissue samples from three individuals on the Visium platform (http://spatial.libd.org/spatialLIBD/). The 12 samples measured on an average of 3973 spots that were manually annotated to one of the six prefrontal cortex layers or white matter. We used sample 151676 as the main analysis example, which contains expression measurement of 33,538 genes on 3460 spots. We presented the results for the other 11 samples in the Supplementary Figures. In the analysis, we retained genes with the non-zero expression on at least 20 spots and retained spots with non-zero expression for at least 20 genes. These filtering steps lead to a final set of 14,690 genes measured on 3460 spots for analysis. The annotated spatial domains in sample 151676 in the

original study is based on cytoarchitecture and include Layer 1 ($n = 289$), Layer 2 ($n = 254$), Layer 3 ($n = 836$), Layer 4 ($n = 254$), Layer 5 ($n = 649$), Layer 6 ($n = 616$), white matter ($n = 533$), and 29 undetermined locations. In the spatial domain detection analysis described in the following section, we excluded the undetermined locations and treated the remaining regional annotations as ground truth.

**Mouse cerebellum data by Slide-seq.** We obtained the Slide-seq data on mouse cerebellum from the Broad Single Cell Portal (ID SCP354)[1]. We used the file "Puck_180430_6", which contains 18,671 genes measured on 25,551 spatial locations. We removed mitochondrial genes and retained genes with non-zero expression level on at least 20 locations. We also retained locations with non-zero expression for at least 20 genes. These filtering steps lead to a final set of 10,515 genes on 20,982 locations for analysis. The Slide-seq data does not come with tissue domain annotations that can serve as ground truth as tissue domain detection was not performed in the original study. Therefore, we relied on the Allen Brain Atlas[108] and other literature[53,54,109–116] to help determine which cluster corresponds to which tissue structure. Because of a lack of corresponding H&E image, such annotations are approximate and cannot be used as ground truth to evaluate spatial clustering performance.

**Mouse hippocampus data by Slide-seq V2.** We obtained the Slide-seq V2 data[2] from https://singlecell.broadinstitute.org/single_cell/study/SCP815/sensitive-spatial-genome-wide-expression-profiling-at-cellular-resolution#study-summary. We used the file "Puck_200115_08", which contains 23,264 genes measured on 53,208 spatial locations. We retained genes with non-zero expression level on at least 20 locations and locations with non-zero expression for at least 20 genes. These filtering steps lead to a final set of 16,235 genes on 51,398 locations for analysis. The Slide-seq V2 data does not come with tissue domain annotations that can serve as ground truth as tissue domain detection was not performed in the original study. Therefore, we relied on the Allen Brain Atlas[108] and other literature[117–120] to help determine which cluster corresponds to which tissue structure. Because of a lack of corresponding H&E image, such annotations are approximate and cannot be used as ground truth to evaluate spatial clustering performance.

**HER2 tumor data by spatial transcriptomics (ST).** We downloaded the HER2-positive breast tumor data collected from ST platform from https://github.com/almaan/her2st (Andersson et al. 2021[58]). We used sample H1 as the main analysis example, which contains expression measurements of 15,030 genes on 613 spatial locations. The results for the other 7 annotated samples are provided in the Supplementary Figures. In the analysis, we retained genes with non-zero expression on at least 20 spots and retained spots with non-zero expression for at least 20 genes. We also removed 21 genes associated with a ring pattern observed in multiple samples in the original study. These genes were confounded by technical artifacts, as explained in the original study. These filtering steps lead to a final set of 10,053 genes measured on 607 spots for analysis. The ST data consists of seven spatial domains that were annotated by pathologists based on H&E staining of the same tissue section. The seven annotated spatial domains include in situ cancer ($n = 97$ spatial locations), invasive cancer ($n = 90$), breast glands ($n = 39$), adipose tissue ($n = 112$), immune infiltrate ($n = 23$), connective tissue ($n = 166$), and other spots in the undetermined region ($n = 80$). In the spatial domain detection analysis described in the following section, we excluded the undetermined region and treated the remaining regional annotations as ground truth.

### Analysis details
**Data normalization and dimension reduction.** For all datasets, we normalized the raw expression count matrix using the variance

stabilizing transformation (VST) implemented in the Seurat function in SCTransform[97]. We then standardized the normalized expression values of each gene to have zero mean and unit standard deviation following[88,96,97]. We used the resulting standardized expression matrix as input for dimension reduction methods that include SpatialPCA and PCA, and used standardized log normalized gene expression with the same set of genes as input for NMF.

For the three spatial domain detection methods, we directly used their software default for gene filtering. For the three dimension reduction methods, we performed gene filtering to retain a set of spatially expressed genes as input. To do so, we followed the recommendation of SPARK-X[82] to use SPARK[77,121] for SVG analysis in small datasets (ST tumor and DLPFC) due to its higher statistical power and use SPARK-X for SVG analysis in larger datasets (Slide-seq and Slide-seq V2) due to its computational efficiency. Genes are declared as significant based on an FDR threshold of 0.05 using the Benjamini–Yekutieli procedure implemented in the software. We used up to 3000 significant SVGs as input, with 2720, 319, 787, and 3000 SVGs selected in the DLPFC, ST tumor, Slide-seq, and Slide-seq V2 data, respectively. We applied SpatialPCA and PCA on the same standardized data matrix for the selected SVGs. In particular, we fitted PCA by applying singular value decomposition (SVD)[122]. For NMF, we first performed log normalization on the gene expression count matrix for the selected spatially expressed genes using the logNormCounts function in scater. Afterward, we used the calculateNMF function in scater[123] to perform NMF and obtain low-dimensional components. Regardless of the method, we extracted the top 20 low-dimensional components for downstream analyses.

We examined the sensitivity of different methods by carrying out dimension reduction using three different sets of genes as input in the DLPFC data. The three gene sets include all genes, 3000 top highly variable genes (HVGs), and 3000 top SVGs. The HVGs are obtained using the FindVariableGenes function in the Seurat package. For the set of all genes, applying SpatialPCA in DLPFC incurs a heavy computational burden due to its iterative eigen decomposition of a gene-by-gene matrix. Therefore, we first performed PCA to extract the top 3000 regular PCs from the set of all genes and then used them as input for SpatialPCA.

We performed additional analysis in the DLPFC dataset using a kernel computed based on a non-Euclidean Delaunay triangulation-based distance. The Delaunay triangulation of the spatial locations on the tissue section is equivalent to the Voronoi diagram for the same set of locations[124]. Therefore, we first calculated the Voronoi tessellation of all locations and defined pairs of locations as Delaunay neighbors if they share a common edge in the Voronoi polygons to construct the Delaunay triangulation. Afterward, we used the delaunayDistance function in the R package spatstat.geom to calculate the distance between each pair of locations $i$ and $j$ in the Delaunay triangulation, which is the minimum number of edges one must travel through between the two locations. We then converted the calculated Delaunay distance to construct the covariance matrix used in SpatialPCA.

**RGB plots for visualizing low-dimensional components.** For SpatialPCA, PCA, and NMF, we summarized the inferred low-dimensional components into three UMAP or tSNE components and visualized the three resulting components with red/green/blue (RGB) colors through RGB plot. For each method, we normalized the tSNE or UMAP embeddings to be in the range of 0 to 1 and used these normalized values as input for the rgb() function in the ggplot function to generate RGB plot. In the process, we explored the sensitivity of RGB plot to the scale of the input low-dimensional components by multiplying them 10 or 20 times. On the RGB plot, we also summarized the RGB colors on each location into a vector in the form of weighted RGB values $RGB_{weighted} = \frac{R^*var(R) + G^*var(G) + B^*var(B)}{var(R) + var(G) + var(B)}$, where $R, G, B$ represent vectors of the color values for the three channels; while $var(\bullet)$ represents the

sample variance computed across locations and serves as the weights. We scaled $RGB_{weighted}$ to have mean zero and variance one. In the DLPFC data, we examined the spatial distribution of the weighted RGB values across spatial domains. We also calculated for each spot the variance of the rescaled $RGB_{weighted}$ on the spot and its six nearest spots, where a small variance indicates spatial smoothness in neighborhood RGB values.

**Spatial domain detection via clustering.** We performed clustering analysis on the low-dimensional components to cluster spatial locations. Clustering locations allows us to effectively segment the tissue into distinct spatial domains. Here, we applied two clustering algorithms: the Walktrap method and the Louvain method. For the Walktrap method, we first constructed a shared nearest neighbors (SNN) graph[125] based on the low-dimensional components using the scran package[126]. SNN measures similarity between each pair of locations by counting the number of neighboring locations that are connected to both locations. With the SNN graph, we applied the Walktrap method using the bluster R package[127] to obtain the cluster labels for locations. For the Louvain method, we first built a k-nearest neighbors (KNN) network among locations using the FNN package[128]. In the KNN network, each location is connected to its K-nearest locations in the Euclidean space. With the KNN network, we applied the Louvain community detection algorithm to cluster locations using the igraph R package[129]. We applied the Walktrap method to the small datasets (DLPFC and ST tumor) and due to its heavy computational burden applied the Louvain method to the other two large datasets. We set the number of spatial clusters in the DLPFC and ST tumor datasets based on the ground truth annotation and set the number of clusters to achieve the highest average Silhouette width[130] in the Slide-seq data. In Slide-seq V2, we were unable to calculate the Silhouette width because the required distance matrix is too big due to the large sample size. Therefore, we set the number of clusters in Slide-seq V2 based on the number of spatial domains annotated in Allen Brain Atlas[108]. After clustering, we performed an additional refinement step following SpaGCN to relabel a spot based on the majority of the domain labels from its surrounding spots along with itself if the two are different (four surrounding spots in the ST data and six in the Visium data).

**Extension of SpatialPCA for joint modeling of multiple samples.** We developed an extension of SpatialPCA to jointly model multiple tissue sections, which is common in many spatial transcriptomics datasets[131]. In the extension, we used the IntegrateData function in Seurat to first remove the batch effects from each dataset and obtain an integrated gene expression matrix, so that the analyzed datasets are compatible to each other and are placed on the same manifold. Afterward, we took advantage of the fact that distinct tissue sections are not near each other in physical space and constructed the covariance matrix for the latent factors in the form of a block diagonal matrix: it consists of the kernel matrices constructed using the spatial location information within each dataset, with zero correlation for pairs of locations across datasets. This way, the latent factors within each dataset are correlated a priori across spatial locations while the latent factors across datasets are not correlated a priori. Certainly, if one wants to model the a priori correlation between latent factors across datasets, due to, for example, their similarity in the features extracted from histology images, then one can also modify the kernel matrices by constructing them using features other than spatial location information. We applied the extension of SpatialPCA to analyze the DLPFC data, which consists of tissue sections from three individuals. For this data application, we jointly analyzed three tissue sections, with one section from each individual, using the extension of SpatialPCA.

**Compared methods for spatial domain detection.** We compared the performance of the above three dimension reduction methods with

four spatial domain detection methods for detecting spatial domains on the tissue. The four examined domain detection methods include BayesSpace, SpaGCN, HMRF, and stLearn.

BayesSpace is designed for spatial domain detection in ST or Visium data from 10x genomics. BayesSpace performs PCA on the top 2000 HVGs and models the top 15 PCs to infer the latent cluster labels for spatial domains. BayesSpace can enhance the resolution of clustering in ST or Visium data by segmenting each spot to six (for Visium) or nine subspots (for ST) and uses a Potts prior to infer the spatial cluster labels on the subspots. For fitting BayesSpace, we set the parameter nrep in spatialPreprocess to be 50,000 in DLPFC and Slide-seq data, and to be 10,000 in ST tumor data due to its small sample size following the BayesSpace tutorial. We set the number of burn in to be 1000 in all datasets. We did not run BayesSpace on the Slide-seq V2 data because it requires too much memory (>100 Gb).

SpaGCN[28] uses gene expression, spatial location and histology information as inputs to infer spatial domains on the tissue. Specifically, SpaGCN reduces the dimension of the preprocessed gene expression matrix using PCA to obtain the top 50 PCs as input. SpaGCN builds a graph to incorporate the spatial and histology information, and then uses a graph convolutional layer to reconstruct the gene expression information by aggregating spatial and histological information. In the DLPFC data, for all twelve samples, we followed the SpaGCN tutorial and provided high-resolution H&E image to SpaGCN. In ST tumor data, we used H&E image with lower resolution downloaded from https://github.com/almaan/her2st. For Slide-seq and Slide-seq V2 data, no H&E images is available, so we choose the "histology=False" option in SpaGCN.

HMRF is a method initially designed for image segmentation[33] and has been recently adapted for spatial domain detection[132] in spatial transcriptomics. We fitted HMRF using the Giotto package with default settings. The default setting of HMRF explores 50 different beta values that range from 0 to 98, where the beta value characterizes the smoothness of the detected tissue region boundaries. Because the results for different beta values were very similar in DLPFC, Slide-seq and ST data, we simply report the results based on beta = 10 throughout the text. We did not run HMRF on the Slide-seq V2 data because it requires too much memory (>100 Gb).

stLearn[29] leverages spatial location and histology information to smooth gene expression for downstream clustering and trajectory inference. stLearn uses gene expression, spatial location and histology information as inputs. We followed the default settings in the tutorial of stLearn to perform clustering analysis on the 12 samples in DLPFC. We did not apply stLearn to ST tumor data as the original data paper does not provide a JSON file that is necessary for stLearn. We also did not apply stLearn to the Slide-seq and Slide-seq V2 data as neither has H&E information. In trajectory inference, stLearn applies a diffusion pseudo-time method on the smoothed gene expression data and focuses on ordering the spatial clusters based on their average pseudo-time within clusters. We compared the pseudo-time calculated from stLearn with that from SpatialPCA in sample 151676 of the DLPFC data.

**Spatial clustering performance evaluation.** For the DLPFC and ST tumor datasets that come with spatial domain annotations that can serve as ground truth, we evaluated the performance of different methods in spatial domain detection by comparing the detected spatial domains with truth in two different ways. First, we used standard clustering evaluation metrics that include the adjusted rand index (ARI)[133] and normalized mutual information (NMI)[134]. Second, we directly evaluated the information contained in the output from different methods in predicting the true spatial domains without performing explicit clustering. Specifically, we treated the true spatial domains as the outcome and fitted a multinomial regression model. For the dimension reduction methods, we treated the extracted low-dimensional components as predictors in the regression model. In the

regression model, we computed the McFadden-adjusted pseudo-$R^2$ (see ref. 37) to evaluate the predictive ability of the predictor variables in predicting the ground truth. A higher pseudo-$R^2$ suggests that the method is capable of extracting informative output in predicting the true spatial domains.

In the ST tumor dataset, we also performed an alternative set of fine-grained analysis to measure the clustering accuracy of different spatial domain detection methods. Unlike the DLPFC data that annotated cortical layers based on cytoarchitecture and selected gene markers, the ST tumor data only contains pathologist annotations that are purely based on the H&E image, which may not contain fine-grained transcriptomic information. To compare the performance of different methods in detecting fine-grained details in the ST data, we reasoned that gene expression would capture fine-grained details for the spatial domains on the tissue much better than the high-level H&E image annotations can and thus can serve as an important benchmark for method comparison. Therefore, for each spatial domain in the annotated eight tissue samples in turn, we first performed differential expression (DE) analysis using MAST implemented in the FindMarkers function in Seurat to detect domain-specific DE genes that are enriched in the spatial domain versus the other domains (log2 fold change > 0; adjusted $P < 0.001$). The number of detected DE genes ranges from 19 to 408 (48 DE genes in adipose tissue, 39 in breast glands, 105 in cancer in situ, 53 in connective tissue, 154 in immune infiltrate, 408 in invasive cancer, and 19 in the undetermined region). Next, for each spatial domain in turn, we obtained the averaged expression level of its domain-specific DE genes to serve as the domain-specific metagene. We further scaled the expression values of each metagene to 0–1. The expression of the metagene for each domain likely contains the fine-grained transcriptomic information and can thus be used to evaluate the fine-grained performance of different spatial domain detection methods. Therefore, we examined the enrichment of each of the seven metagenes in the spatial domains detected by each method for method evaluation. Intuitively, if the spatial domain detection method is better powered to detect fine-grained spatial domains, then the expression level of the metagenes would be enriched in one of the detected spatial domains. Therefore, for each method in turn, we first selected the spatial domain with the highest percentage (>=35%) of overlapped spots with the ground truth domain to serve as the annotated domain for the particular method. We then calculated a ratio of mean metagene expression in the domain versus that outside the domain as the metagene enrichment score.

For all datasets, we used another three metrics to evaluate clustering performance. First, we adopted the integration quality quantification metric used in scRNA-seq and calculated the local inverse Simpson's index (LISI)[135] to quantify the clustering performance for spatial domain detection. The LISI score is the effective number of spatial domain labels represented in the local spatial neighborhood of a spot and is calculated as

$$S = \frac{1}{\sum_{k=1}^{K} p(k)} \quad (5)$$

where $p(k)$ is the probability that the spatial domain cluster label $k$ is in the local neighborhood, and $K$ is the total number of spatial domains. We calculated the LISI score using the compute_lisi function from the LISI R package with default parameters (perplexity = 30). Lower LISI score indicates more homogeneous neighborhood spatial domain clusters of the spot.

Second, we adopted image segmentation performance quantification metrics in mass spectrometry imaging[136] and used spatial chaos score (CHAOS) and percentage of abnormal spots (PAS) score to quantify the clustering performance for spatial domains.

The CHAOS score measures the spatial continuity of the detected spatial domains[136,137]. To calculate CHAOS, we first create a one-nearest-

neighbor (1NN) graph for the spots in each spatial cluster by connecting each spot with its nearest neighbor. We denote $w_{ij}$ as the edge weight between spot $i$ and spot $j$, which is calculated as

$$w_{kij} = \begin{cases} d_{ij}, & \text{if spot } i \text{ and } j \text{ are connected in the 1NN graph in cluster } k \\ 0, & \text{otherwise} \end{cases}$$

where $d_{ij}$ is the Euclidean distance between the two spots. We calculate CHAOS as the mean length of the graph edges in the 1NN graph in the following form

$$CHAOS = \frac{\sum_{k=1}^{K} \sum_{i,j}^{n_k} w_{kij}}{N} \qquad (6)$$

where $N$ is the total number of spots; $K$ is the total number of spatial domains; and $n_k$ is the number of spots in $k$-th spatial domain. A smaller CHAOS score reflects better spatial continuity.

The PAS score measures the randomness of the spots that are located outside of the spatial domain where it was clustered to. The PAS score is calculated as the proportion of spots with a cluster label that is different from at least six of its neighboring ten spots. A small PAS score indicates spot homogeneity within spatial clusters.

**Cell-type deconvolution.** We performed deconvolution using the software RCTD with default settings[138]. Cell-type deconvolution in spatial transcriptomics allows us to estimate the cell-type composition on each spatial location using a reference scRNA-seq data.

For the DLPFC data, we used the same scRNA-seq data that we used as a reference in simulations[106] for deconvolution. The scRNA-seq reference data contains expression measurements for 24,153 genes and 2394 cells that belong to six cell types annotated in the original study. The six cell types include astrocytes ($n = 76$ cells), GABAergic neurons ($n = 701$), microglia ($n = 68$), neurons ($n = 1057$), oligodendrocyte progenitor cell (OPC; $n = 117$), and stem cells ($n = 290$).

For the Slide-Seq data, we selected a scRNA-seq data collected on the same tissue, the mouse cerebellum, via Drop-seq[139] to serve as the reference data for deconvolution. The reference data contains 26,139 cells that belong to multiple cell types characterized in the original study. We focus our analysis on major cell types that include granule cell ($n = 21,331$ cells), Purkinje neuron ($n = 178$), interneuron ($n = 1547$), oligodendrocyte ($n = 347$), astrocyte ($n = 326$), Bergmann glia ($n = 1404$), microglia ($n = 64$), choroid plexus ($n = 50$), endothelial cell ($n = 422$), fibroblast-like cell ($n = 241$), and interneurons_and_Other_Nnat ($n = 229$).

For the Slide-seq V2 data, we selected a scRNA-seq data collected on the same tissue, the mouse hippocampus, via Drop-seq[139], to serve as the reference data for deconvolution. The reference data contains 53,204 cells that belong to multiple cell types characterized in the original study. The cell types include Anterior Subiculum, proximal to CA1 ($n = 1556$ cells), Astrocyte ($n = 7503$), CA1 Principal cells ($n = 2489$), CA1 Principal cells (Anterior) ($n = 3088$), CA2 Principal cells ($n = 330$), CA3 Principal cells ($n = 5623$), Cajal-retzius ($n = 336$), Choroid_Plexus ($n = 22$), Deep layer subiculum ($n = 509$), Dentate hilum ($n = 530$), Dentate Principal cells ($n = 13,265$), Endothelial_Stalk ($n = 1903$), Endothelial_Tip ($n = 296$), Entorhinal cortex ($n = 1580$), Entorhinal cortex (IEG) ($n = 27$), Ependymal ($n = 337$), Interneuron ($n = 3035$), Lateral CA3 Principal cells ($n = 649$), Medial entorrhinal cortex ($n = 867$), Medial entorrhinal cortexm ($n = 56$), Microglia ($n = 467$), Mural ($n = 705$), MyelinProcesses ($n = 227$), Neurogenesis (SGZ) ($n = 458$), Neuron ($n = 3087$), Oligodendrocyte ($n = 1969$), Polydendrocyte ($n = 1046$), Postsubiculum ($n = 43$), Resident macrophage ($n = 78$), and Subiculum ($n = 1123$).

For the ST data, we selected a scRNA-seq data collected on breast cancer to serve as the reference data for deconvolution. The reference data is collected on breast cancer via the InDrop platform[140]. It contains 24,271 cells that belong to multiple immune cell types and malignant cell types characterized in the original study. These cell types include B_Cells ($n = 1245$ cells), CD4+ T cells ($n = 2003$), CD8+ T cells ($n = 3691$), Myeloid ($n = 4606$), Myoepithelial ($n = 212$), NK cells ($n = 358$), NKT cells ($n = 164$), Plasma_Cells ($n = 1955$), T_cells_unassigned ($n = 938$), T cells Cycling (proliferating T cells, $n = 605$), T-Regs ($n = 994$), Tfh cells (T-follicular helper, $n = 175$), dPVL (differentiated-perivascular-like, $n = 214$), iCAFs (inflammatory cancer-associated fibroblast, $n = 1129$), imPVL (immature-PVL, $n = 106$), myCAFs (myofibroblast-like CAF, $n = 280$), Endothelial ($n = 610$), Epithelial_Basal (cancer cells, $n = 4095$), Epithelial_Basal_Cycling (cancer cells with high proliferation, $n = 614$), and Epithelial_Luminal_Mature (epithelial normal luminal, $n = 277$). With the scRNA-seq reference data, we estimated the cell-type composition of the reference cell types on each spatial location.

**High-resolution spatial map construction and gene expression prediction.** We constructed a high-resolution spatial map using SpatialPCA and illustrated this feature for the ST tumor data with relatively low resolution. For high-resolution map construction, we followed BayesSpace to impute nine subspots for each ST spot and six subspots for each Visium spot, as the ST and Visium spots are arranged on square and hexagonal lattices. For the other technologies, we simply set the default imputation choice to be imputing on four new locations for each measured location. In particular, we scaled the $x$- and $y$ coordinates of the measured spatial locations to have mean zero and unit standard deviation. We then replaced each measured spatial location with four new locations by adding or subtracting the same small distance on both x and y coordinates of the measured location. Subsequently, the four new locations form a square, with the original measured location sitting in the center. The small distance used to create the new locations is data dependent: it is calculated as ¼ of the median distance between one location with its nearest neighbor. Afterward, we imputed spatial PCs and the expression level of individual genes on the new locations. Note that spatial imputation is only performed on points in space that are surrounded by observed expression. Extrapolating outside the measured tissue area would be challenging, especially when the extrapolated area contains distinct cell-type composition and transcriptomic profiles as compared to the measured area.

**Trajectory inference on the tissue.** Besides spatial domain detection, we also applied Slingshot[19] on the low-dimensional components to infer the developmental trajectories among spatial locations on the tissue in the DLPFC, ST tumor, and Slide-seq V2 data[83,84]. Slingshot is a common scRNA-seq trajectory inference method originally applied on regular PCs. Here, we applied slingshot on the spatial PCs so that the nearby tissue locations with similar gene expression will be inferred with similar pseudo-time. Slingshot requires the cluster labels as additional input. We used the cluster labels obtained from the clustering analysis described in a previous subsection for Slingshot. After trajectory inference, Slingshot assigned each location a pseudo-time value. Slingshot requires users to specify the start of the trajectory (i.e., the beginning of pseudo-time) based on biological knowledge. In the trajectory analysis, almost all lineage inference methods[21,30;] require the user to designate one cluster as the start cluster, or one cell as the start cell. Setting a start cluster only changes the direction of the trajectory but does not influence the relative position of clusters on the trajectory. We set the white matter as the start cluster for DLPFC, set the tumor region as the start cluster for ST, and set the layer 6 as start cluster for Slide-seq V2. In Slide-seq V2 data, we focused the trajectory inference only on the cortical layers 4, 5, and 6, because slingshot cannot handle the large sample size on the whole tissue slice and because the cortical layers are naturally ordered as they are developed in a sequential fashion during corticogenesis. In ST data, we focused on trajectory inference on tumor and tumor-adjacent regions to investigate how these locations are connected to one another and underlie tumorigenesis. Based on the inferred pseudo-time values, we

connected neighboring locations on the tissue to construct trajectories. Specifically, we first overlayed the tissue in each data with an evenly spaced square grid graph. For each smallest square on the grip graph, we obtained inside the square the two locations that have the largest and smallest pseudo-time values. We then draw an arrow line connecting these two locations, with the arrow pointing towards the location with the larger pseudo-time. The size of the grid graph is data dependent: we searched in sequence a set of pre-determined numbers (10, 15, 20, 25, 30) as grid size choice for each dataset, and selected among them the smallest number that ensures at least 1/3 of the arrow lines to start from one spatial domain and end at another spatial domain. This way, an appreciable fraction of arrow lines would cross regional boundaries, ensuring effective visualization of trajectories between spatial domains. With the strategy, we used a 10 by 10 grid for the ST data and a 20 by 20 grid for the DLPFC data. We also applied stLearn on DLPFC sample 151676 to infer the trajectory and we set the spot on the left-bottom corner of white matter as the root spot in the inference. We did not apply stLearn to the other datasets due to a lack of H&E image in Slide-seq and Slide-seq V2 data, and a lack of JSON file in the HER2-positive breast tumor data.

**Differential gene expression analysis and enrichment analysis.** We performed differential gene expression (DE) analyses and subsequent gene set enrichment analyses on all of four datasets. We performed three sets of DE analyses. The first set of DE analyses aims to identify spatial domain-specific DE genes. In particular, we examined the SpatialPCA detected spatial domains one at a time and used MAST[141] wrapped in the FindMarkers function in the Seurat package to identify genes that are DE in the domain of focus as compared to all other domains at a false discovery rate (FDR) of 0.05. The second set of DE analyses aims to identify pseudo-time-associated genes. In particular, we used Pearson's correlation test to identify genes that were associated with the inferred pseudo-time based on a Bonferroni corrected p value threshold of 0.05. The third set of DE analyses aims to identify genes associated with spatial PCs. In particular, we used Pearson's correlation test to identify genes that were associated with the inferred spatial PCs based on a Bonferroni corrected p value threshold of 0.05. After each set of DE analyses, we performed gene set enrichment analyses (GSEA) on the detected DE genes using the g:GOSt function in gProfiler2 package. In the enrichment analysis, we used all expressed genes as background[142] and used the default option g:SCS method in gProfiler2 for multiple testing correction. The gene sets are downloaded from the Molecular Signatures Database (MSigDB[143,144]) available from the Broad Institute, including the C2 (KEGG), C5 (GO BP: biological processes, GO CC: cellular components, GO MF: molecular functions and HP:Human Phenotype Ontology) and hallmark pathway datasets. For the ST tumor data, we also included cancer-related gene sets, including C4 (cancer modules) and C7 (immunologic signatures). For the Slide-seq cerebellum data, we converted the MGI gene IDs to human homolog gene symbols before GSEA analysis[145].

#### Reporting summary
Further information on research design is available in the Nature Portfolio Reporting Summary linked to this article.

## Data availability
This study made use of publicly available datasets. The human DLPFC samples[35] are available at http://spatial.libd.org/spatialLIBD/. ST data[58] are available at https://github.com/almaan/her2st. Slide-Seq data[1] are available at Broad Institute's single-cell repository (https://singlecell.broadinstitute.org/single_cell/) with ID SCP354. Slide-seq V2 data[2] are available at Broad Institute's single-cell repository (https://singlecell.broadinstitute.org/single_cell/) with ID SCP815. The scRNA-seq reference data used in this study are all publicly available, including GSE104276[106] for human prefrontal cortex; http://dropviz.org[139] for

mouse cerebellum and hippocampus; and GSE114725[140] for human breast tumor. The databases we used include Molecular Signatures Database[143,144] (https://www.gsea-msigdb.org/gsea/msigdb), Mouse Genome Database[145] (http://www.informatics.jax.org), and Allen Brain Atlas[108] (https://www.brain-map.org). Source data are provided with this paper.

## Code availability
The SpatialPCA software code[146] is publicly available at http://xzlab.org/software.html and https://github.com/shangll123/SpatialPCA. The source code is released under the GNU General Public License version 3 (GPL >=3). Example codes for using SpatialPCA are publicly available at http://lulushang.org/SpatialPCA_Tutorial/index.html. All analysis codes for reproducing the results of the present study are publicly available at https://github.com/shangll123/SpatialPCA_analysis_codes.

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

## Acknowledgements

This study was supported by the National Institutes of Health (NIH) Grants R01GM126553, R01HG011883, and R01GM144960 (all to X.Z.).

## Author contributions

X.Z. conceived the idea and provided funding support. L.S. and X.Z. designed the experiments. L.S. developed the method, implemented the software, performed simulations, and analyzed real data. L.S. and X.Z. wrote the manuscript.

## Competing interests

The authors declare no competing interests.
