## [Peer Review File · Nature Communications]

REVIEWER COMMENTS

Reviewer #1 (Remarks to the Author): Expert in spatial transcriptomics and bioinformatics

Shang and Zhou present their method, SpatialPCA, that performs spatially aware dimensionality reduction, which can be used for further downstream analytical tasks common to analysing spatial transcriptomics data. SpatialPCA is a probabilistic factor model that incorporates a Gaussian kernel to relate nearby spatial transcriptomics spots with each other, with the idea that cells near each other in physical space are expected to be similar in their gene expression profiles and local cell type proportion identities. The SpatialPCA manuscript is well written and very straightforward to follow. The simulations are extensive and range of real-data explorations are sufficient, in my opinion, to show comparable or superior performance. The authors provide software on Github and all scripts for reproducing the analyses are online. Nevertheless, I have some comments that I think, if addressed, would result in an improved manuscript.

Comments:

- It's unclear whether SpatialPCA can be performed for replicated spatial transcriptomics datasets. That is, if multiple distinct tissues are captured which are not near each other in physical space, how would one go about using SpatialPCA to embed these data into the same low-dimensional space?

- The SpatialPCA model includes a kernel bandwidth parameter, γ , that indicates the degree of spatial correlation among cells in proximity to each other. While the authors use a "rule of thumb" bandwidth as a reasonable default, I'm curious as to the level of robustness of downstream results (e.g. clustering) when varying this parameter.

- The authors perform comprehensive comparisons against other methods, namely SpaGCN, BayesSpace, NMF and stLearn. For the purposes of spatial clustering, it may be worth comparing to methods that have specifically been designed with this purpose in mind, e.g. the hidden markov random field (HMRF) approach with doi: 10.1038/nbt.4260.

- I would be interested in the performance of SpatialPCA for a non-spot based spatial transcriptomics dataset, e.g. the technologies of ISS/MERFISH/SEQFISH. Does SpatialPCA still outrank other methods if the observed data is more likely to be truly single cell - with the exception of segmentation errors?

- Typical use of PCA for scRNA-seq analysis involves a primary feature selection step, e.g. selecting for highly variable genes (HVGs) prior to further dimensionality reduction. While there is some mention of HVGs in the manuscript, it's unclear to me whether some a priori feature selection step is taken by SpatialPCA other than minimal filtering of detected genes. The authors could clarify this in the Methods or Discussion.

- It may be that the biological relevance of physical coordinates are non-Euclidean, e.g. a long thin tissue sample could be twisted in the process of data capture. What approaches could be taken to incorporate constraints on the spatial covariance matrix, or introduce a different spot-spot or cell-cell distance matrix altogether, for example one that is extracted from the Delaunay triangulation of cells?

- Spatial imputation, it may be worth reiterating that this relates to points in space that are surrounded by observed expression. I would be curious as to what degree of extrapolation outside of the measured tissue area would be possible with SpatialPCA, although I do not think this is essential.

Reviewer #2 (Remarks to the Author): Expert in spatial transcriptomics and bioinformatics

The authors present a novel computational method for dimensionality reduction of variance-stabilized and feature-normalized high-dimensional spatial transcriptomics data, called SpatialPCA. The method builds on an existing factorization method, Probabilistic PCA, by estimating a covariance matrix for the Multivariable Gaussian prior distribution for latent factors. This choice encourages the latent factors to be spatially smooth. Below I summarize my remarks and concerns about the evaluation of the method.

The method promises to deliver:

- Enriched biological signal, however this is not objectively evaluated;
- Preserved spatial correlation;
- Ability to impute the low-dimensional components at unmeasured arbitrary spatial locations enabling increasing the resolution to be arbitrarily higher than the measured one. Perhaps the SpatialPCA generative model is more reliable in interpolation rather than extrapolation regime, meaning predicting the expression over locations surrounded by observed locations. Even in this interpolation regime, the latent factors are continuous over space, and it remains the user's discretion how to appropriately choose the density of spots they want to impute.

The output of the SpatialPCA method are latent factors which can be used as an input to downstream existing methods. The authors have paired their spatial PCs with three types of downstream methods to address the following questions:

- Spatial transcriptomic visualization - this is done by 1. Dim red via SpatialPCA, 2. More dim red (umap or tsne) to three dimensions, 3. assigning RGB to each of the three axes, 4. Plot the colors.

- Spatial domain detection - pair SpatialPCA with existing clustering method - the method seems to be good at detecting spatially contiguous and smoothly varying spatial domains - which is not always a realistic assumption, for example in the presence of a stripe pattern.

- Spatial trajectory inference - they pair SpatialPCA with an existing trajectory inference method. Single-cell RNA-seq trajectory inference methods are designed to be applied to single cell resolution data. It is unclear how reliable such methods are for larger/mixed spatial transcriptomics spots.

First, the performance of the SpatialPCA method is evaluated through simulations and comparisons to other methods. The quality of the factors derived from SpatialPCA is evaluated in the context of one of the downstream tasks – domain detection. The authors simulate a simplified layered tissue consisting of 4 layers, 10k single cell locations from 4 cell types, across 5000 genes. To match the lack of single-cell resolution of many spatial transcriptomics technologies, the authors perform spot aggregation at different resolutions. The counts for simulated cells are binned, and not ever “split” between neighboring spots, which is likely to happen in technologies such as Slide-seq and ST or Visium, for example. This sharing of counts among neighboring spots will lead to artifactually higher spatial correlation between spots in real data, and is not well represented in the simulation design.

Finally, the authors apply their method, SpatialPCA, paired with existing downstream analysis methods, to four public spatial transcriptomics datasets from different technologies and different tissue structures: cortex, mouse hippocampus, and tumor microenvironment.

One of the methods chosen as a benchmark is NMF, i.e. Non-negative matrix factorization. As the name implies, the method is designed to be applied to non-negative data. However, the authors apply it to data after normalization as described in the Methods sections, when the data is no longer non-negative. This makes the comparisons to NMF inappropriate.

A key metric used by the authors in the evaluation of the ability of SpatialPCA to predict pathologist annotations is McFadden’s pseudo R^2 . This metric is not described in the methods and it remains unclear to this reviewer how it can be applied to PCA or NMF, as they are not identified with a likelihood.

The method is reported to not be sensitive to choice of hyperparameters, such as number of factors. However, the evaluation is purely visual and subjective (see Figures S5, S10, S14, S19).

The work has potential to be useful to researchers seeking to identify spatial domains denovo. It should be pointed out that the work bares a large degree of similarity with:

<https://doi.org/10.48550/arXiv.2110.06122> Nonnegative spatial factorization by F. William Townes, Barbara E. Engelhardt

<https://doi.org/10.1038/s41592-021-01343-9> Identifying temporal and spatial patterns of variation from multimodal data using MEFISTO by Velten et al.

Reviewer #3 (Remarks to the Author): Expert in bioinformatics, cancer genomics, immunology, and tumour microenvironment

In this study, the authors present a dimensionality reduction approach tailored for spatial transcriptomics. The authors show the new methods utility across simulations and multiple datasets and compared to other methods. Altogether, this is an impressive piece of work in an up-and-coming field of research. The method seems sound and there is a lot of work here showing its usability to extract insights.

I focus here on comments on the HER2 tumor data analysis

1. The original paper provides data on 36 sections, and there are manual annotations in 6 of them. The analysis shown here focuses on one sample only, and shows marginal advantage for SpatialPCA over other methods, and can be random. I think that it would be much more powerful to show statistics across all samples.

2. From the text it is suggested that there is very high matching between the expert annotation and the annotation by SpatialPCA, but by eye it doesn't look that well. For example, DCIS, cancer and fibrous tissue are not distinguished at all. There are many differences. I understand that other methods aren't doing better, but this suggests to me that this whole benchmarking approach – trying to identify major domains – might not make much sense. The domains identified by the pathologist are very “high-level” and there shouldn't be much resemblance to the fine-grained analysis that is possible with ST.

3. “We found that the regional specific genes detected by SpatialPCA are highly enriched in pathway of immune response and biological adhesion” – this sentence is unclear to me.

4. I don’t understand that deconvolution analysis that included only immune cells. This surely creates bias. For example, the mast cell result is most probably some kind of bias (to my best knowledge, there aren’t supposed to be many mast cells). My guess is that since the reference of mast cells have some resemblance to stroma cells, it is “enriched” in this analysis.

5. “In PCA and NMF, the detected tumor regions are not as smooth and continuous as SpatialPCA.” – of course, SpatialPCA underlying hypothesis is of smoothness, and the method inherently pushes for that.

6. “with multiple unique features of TLS: the region is located near the tumor; the region primarily contains T cells and B cells, all of which are key cell types in TLS” – those are not unique features of TLS, but for almost any region enriched with immune cells in the TME. I am not an expert, but there are more markers required to determine a region as TLS. See Box 1 here <https://www.nature.com/articles/s41568-019-0144-6>. Since it is possible to identify TLS with H&E, I think it would be best to compare predictions of TLS to an expert determination.

7. The identification of the surrounding region is interesting, but I think there are too many speculations here for a results section, and most of the text here should be moved to the discussion.

Minor comments:

BayesSpace seems to be better in some of the statistics (PAS and CHAOS). It should be noted.

Supp figure are not in order (S30, S29)

Figure S18 –

a. Missing axes

b. Measuring variance doesn’t seem like a fair approach. SpatialPCA is actively smoothing the data, so variance is expected to be low.

c. Not clear what is the first row.

d. Why no comparison to BayesSpace and SpaGCN

5f-h before 5e

REVIEWER COMMENTS

Reviewer #1 (Remarks to the Author): Expert in spatial transcriptomics and bioinformatics

Shang and Zhou present their method, SpatialPCA, that performs spatially aware dimensionality reduction, which can be used for further downstream analytical tasks common to analyzing spatial transcriptomics data. SpatialPCA is a probabilistic factor model that incorporates a Gaussian kernel to relate nearby spatial transcriptomics spots with each other, with the idea that cells near each other in physical space are expected to be similar in their gene expression profiles and local cell type proportion identities. The SpatialPCA manuscript is well written and very straightforward to follow. The simulations are extensive and range of real-data explorations are sufficient, in my opinion, to show comparable or superior performance. The authors provide software on Github and all scripts for reproducing the analyses are online. Nevertheless, I have some comments that I think, if addressed, would result in an improved manuscript.

Thank you for your constructive comments. Our detailed responses are listed below.

Comments:

1. It's unclear whether SpatialPCA can be performed for replicated spatial transcriptomics datasets. That is, if multiple distinct tissues are captured which are not near each other in physical space, how would one go about using SpatialPCA to embed these data into the same low-dimensional space?

Thank you for the comment. The previous version of SpatialPCA cannot handle replicated spatial transcriptomics datasets. Following your comment, we have developed a new extension of SpatialPCA to perform dimension reduction on replicated datasets. In the extension, we take advantage of the fact that distinct tissue sections are not near each other in physical space and construct the covariance matrix for the latent factors in the form of a block diagonal matrix: it consists of the kernel matrices constructed using the spatial location information within each dataset, with zero correlation for pairs of locations across datasets. This way, the latent factors within each dataset are correlated *a priori* across spatial locations while the latent factors across datasets are not correlated *a priori*. (Certainly, if one wants to model the *a priori* correlation between latent factors across datasets, due to, for example, their similarity in the features extracted from histology images, then one can also modify the kernel matrices by constructing them using features other than spatial location information; we did not explore this direction further due to time constraint). In addition to the model extension, prior to analysis, we also used the IntegrateData function in Seurat to first remove the batch effects from each dataset and obtain an integrated gene expression matrix, so that the analyzed datasets are compatible to each other and are placed on the same manifold. We have included these technical details for the SpatialPCA extension in the Methods section (lines 1001-1017 on page 24).

We have applied the extension of SpatialPCA to analyze the DLPFC data, which consists of tissue sections from three individuals. We analyzed one section from each individual using the extension of SpatialPCA and we found that the extension of SpatialPCA can help substantially improve the spatial domain detection accuracy for some tissue sections but at the slight cost of the accuracy loss of other sections. Specifically, the single dataset analysis version of SpatialPCA achieves an ARI of 0.540, 0.376 and 0.577 for samples 151507, 151669 and 151673, respectively; while the extension of SpatialPCA achieves an ARI of 0.518, 0.431 and 0.552. Therefore, the extension of SpatialPCA substantially improves spatial domain detection accuracy on the sample 151669, which has a poor accuracy when analyzed individually, but at the cost of a small reduction in accuracy for the other two samples. The new results are shown in Supplementary Figure S10C and are described in detail in the updated Discussion section (lines 583-590 on page 14).

2. The SpatialPCA model includes a kernel bandwidth parameter, γ , that indicates the degree of spatial correlation among cells in proximity to each other. While the authors use a "rule of thumb" bandwidth as a reasonable default, I'm curious as to the level of robustness of downstream results (e.g. clustering) when varying this parameter.

Thank you for the comment. Following your comment, we have performed additional sensitivity analysis to examine the robustness of results with respect to bandwidth selection in the DLPFC dataset. Let's take the sample 151676 as an example. The bandwidth selected for this tissue section using the "rule of thumb" is 0.0351, which achieves an ARI of 0.635 for spatial domain detection. In the new sensitivity analysis, we varied the bandwidth from 0.02 to 0.05, with an increment of 0.001. In the sensitivity analysis, we found that the spatial clustering accuracy by SpatialPCA are reasonably robust with respect to the bandwidth parameter, with the median ARI being 0.595 (min=0.541, max=0.646) when the bandwidth parameter ranges from 0.02 to 0.05. The new results are shown in Supplementary Figure S10A and are described in the updated Result section (lines 258-259 on page 7).

3. The authors perform comprehensive comparisons against other methods, namely SpaGCN, BayesSpace, NMF and stLearn. For the purposes of spatial clustering, it may be worth comparing to methods that have specifically been designed with this purpose in mind, e.g. the hidden markov random field (HMRF) approach with doi: 10.1038/nbt.4260.

Thank you for the comment. Following your comment, we have included HMRF into comparison in both simulations and in all real data applications except for the Slide-seq V2 data. We were unable to run HMRF on the Slide-seq V2 data because it required too much memory (>100Gb). For all analyses, we fitted HMRF using the Giotto package with default settings[1]. In the analysis, for cell-level simulations, HMRF is ranked as the second-best method right after SpatialPCA for spatial domain detection in three out of four scenarios and is ranked as the 6th method for the remaining scenario. For spot-level simulations, HMRF is ranked as the 3rd method in ten out of twelve scenarios and is ranked as 2nd method for the remaining scenarios. In the real data, HMRF

is ranked as the 5th method in the DLPFC data and 4th in the ST tumor data. The new results are shown in Figure 1-5, Supplementary Figures S1-S5, S11-S12, S19, S22-S23, S26, S29, S35-S37, S39 and S43, Supplementary Tables S1-S5, S7 and S13, and are described in detail in the updated Introduction section (lines 95-97 on page 3), Results section (lines 141-144 on page 4, lines 165-168 on page 5, lines 189-191 on page 5, lines 262 and 267 on page 7, line 313 on page 8, lines 441, 446-447, and 457-459 on page 11) and Methods section (lines 1045-1051 on page 25).

4. I would be interested in the performance of SpatialPCA for a non-spot based spatial transcriptomics dataset, e.g. the technologies of ISS/MERFISH/SEQFISH. Does SpatialPCA still outrank other methods if the observed data is more likely to be truly single cell - with the exception of segmentation errors?

Thank you for the comment. Following your comment, we obtained a MERFISH dataset [2] and applied all methods to this data. The MERFISH data is collected on the preoptic region of the mouse hypothalamus using the multiplexed error-robust fluorescent *in situ* hybridization-based technology. We obtained all five tissue slices in animal1 (bregma values: -0.04, -0.09, -0.14, -0.19, and -0.24), with 155 genes and an average of 5,663 cells. These slices contain ground truth spatial domain annotations, which were manually annotated according to cell type distribution and marker gene expression in [3]. In the analysis, consistent with the other datasets, we found that SpatialPCA outperforms the other methods. Specifically, in terms of spatial clustering accuracy, SpatialPCA achieves a median ARI of 0.454, which is higher than the other methods (BayesSpace 0.1, SpaGCN 0.262, HMRF 0.414, NMF 0.06, PCA 0.07). In terms of spatial continuity of the detected spatial domains, SpatialPCA achieves a median PAS of 0.043, which is also lower than the other methods (BayesSpace 0.732, SpaGCN 0.359, HMRF 0.048, NMF 0.356, PCA 0.369). The new results are shown in Supplementary Figures S43 and are described in detail in the updated Discussion section (lines 590-594 on pages 14).

5. Typical use of PCA for scRNA-seq analysis involves a primary feature selection step, e.g. selecting for highly variable genes (HVGs) prior to further dimensionality reduction. While there is some mention of HVGs in the manuscript, it's unclear to me whether some a priori feature selection step is taken by SpatialPCA other than minimal filtering of detected genes. The authors could clarify this in the Methods or Discussion.

Thank you for the comment. Indeed, we followed standard approaches and selected SVGs prior to dimension reduction for all methods including SpatialPCA. These details were previously described in the feature selection part inside the "Data normalization and dimension reduction" subsection under the "Analysis Details" section in Methods. Following your comment, we have made this clearer to readers in the Methods section (lines 924-926 on page 22).

6. It may be that the biological relevance of physical coordinates are non-Euclidean, e.g. a long thin tissue sample could be twisted in the process of data capture. What approaches could be

taken to incorporate constraints on the spatial covariance matrix, or introduce a different spot-spot or cell-cell distance matrix altogether, for example one that is extracted from the Delaunay triangulation of cells?

Thank you for the comment. The modeling framework of SpatialPCA is general and can be paired with a covariance matrix constructed using non-Euclidean distance. To illustrate such application, we have performed additional analysis in the DLPFC dataset using Delaunay triangulation based distance. The Delaunay triangulation of the spatial locations on the tissue section is equivalent to the Voronoi diagram for the same set of locations [4]. Therefore, we first calculated the Voronoi tessellation of all locations and defined pairs of locations as Delaunay neighbors if they share a common edge in the Voronoi polygons to construct the Delaunay triangulation. Afterwards, we used the `delaunayDistance` function in the R package `spatstat.geom` to calculate the distance between each pair of locations i and j in the Delaunay triangulation, which is the minimum number of edges one must travel through between the two locations. We then converted the calculated Delaunay distance to construct the covariance matrix used in SpatialPCA. Using the Delaunay distance in SpatialPCA, we obtained an ARI of 0.453 for spatial domain detection, which is lower than using the default covariance matrix constructed by Euclidean distance (ARI=0.577). While non-Euclidean distance does not work as well as Euclidean distance in this particular dataset, we acknowledge that non-Euclidean distance could be beneficial in other datasets. Therefore, we have implemented an option in SpatialPCA to compute the covariance matrix based on Delaunay triangulation. The new results are shown in Supplementary Figures S10B and are described in the updated Discussion section (lines 568-576 on page 14) and Methods section (lines 948-957 on page 23).

7. Spatial imputation, it may be worth reiterating that this relates to points in space that are surrounded by observed expression. I would be curious as to what degree of extrapolation outside of the measured tissue area would be possible with SpatialPCA, although I do not think this is essential.

Thank you for the comment. We have included a few sentences in the Methods section to reiterate that the spatial imputation relates to points in space that are surrounded by observed expression (lines 1200-1203 on page 28). Indeed, extrapolating outside the measured tissue area would be challenging especially when the extrapolated area contains distinct cell type composition and transcriptomic profiles as compared to the measured area.

Reviewer #2 (Remarks to the Author): Expert in spatial transcriptomics and bioinformatics

The authors present a novel computational method for dimensionality reduction of variance-stabilized and feature-normalized high-dimensional spatial transcriptomics data, called SpatialPCA. The method builds on an existing factorization method, Probabilistic PCA, by estimating a covariance matrix for the Multivariable Gaussian prior distribution for latent factors. This choice encourages the latent factors to be spatially smooth. Below I summarize my remarks and concerns about the evaluation of the method.

Thank you for your constructive comments. Our detailed responses are listed below.

(1) The method promises to deliver:

- Enriched biological signal, however this is not objectively evaluated;
- Preserved spatial correlation;
- Ability to impute the low-dimensional components at unmeasured arbitrary spatial locations enabling increasing the resolution to be arbitrarily higher than the measured one. Perhaps the SpatialPCA generative model is more reliable in interpolation rather than extrapolation regime, meaning predicting the expression over locations surrounded by observed locations. Even in this interpolation regime, the latent factors are continuous over space, and it remains the user's discretion how to appropriately choose the density of spots they want to impute.

Thank you for the nice summary and comments.

We are a bit lost on your comment on the enriched biological signal. Previously we have performed objective and quantitative evaluations to show that the latent factors from SpatialPCA contain enriched biological signal, by showing that the latent factors can be used to perform multiple downstream analyses, better than the latent factors obtained from other dimension reduction methods. For example, the latent factors from SpatialPCA can help detect spatial domains more accurately than the other methods, with higher ARI and NMI scores as well as lower PAS and CHAOS scores, across datasets. In addition, the latent factors from SpatialPCA can be paired with single cell trajectory inference methods and detected the "inside-out" pattern of corticogenesis more consistently than the other methods. Our approach of objectively evaluating the biology information contained in the latent factors from SpatialPCA follows exactly the long-standing tradition of single cell genomics in evaluating the biological signal contained in low dimensional features through clustering analysis and pseudotime inference [5-8]. We suspect that different scientists may have a different interpretation on the term "Enriched" and our use of "Enriched" may have led to some level of miscommunication. Therefore, we have removed the word "Enriched" from the abstract following your comment and simply state that the latent factors from SpatialPCA contain biological information that can be used for various downstream analysis.

Besides these existing objective and quantitative evaluations, we have included additional qualitative analysis in the revised manuscript to interpret the biological signals contained in the

latent factors. Specifically, for each examined dataset, we identified genes that are associated with each spatial PC and performed gene set enrichment analysis on them to interpret the biological signals contained in the spatial PCs. We found that the pathways enriched in spatial PC associated genes all make biological sense. For example, in DLPFC data, the top spatial PC associated genes are enriched in synapse related, neuron projection, and synaptic signaling pathways. In Slide-Seq data, the top spatial PC related genes are enriched in synapse related, axon related, synaptic signaling, and oligodendrocyte differentiation pathways. In Slide-seq V2 data, the top spatial PC associated genes are enriched in synapse related, cilium movement, electron transport and neurogenesis pathways. In ST tumor data, the top spatial PC related genes are enriched in cell activation, cell-cell adhesion, cell migration, and immune response pathways. The new results are shown in Supplementary Figures S8, S18, S25 and S33, with details provided in the Results section (lines 252-254 on page 6, lines 302-303 on page 8, lines 370-371 on page 9, lines 424-426 on page 10) and Methods section (lines 1254-1256 on page 29).

We fully agree with your comments on imputation. Following your comments, we have included a few sentences in the Methods section to reiterate that, just like any other spatial imputation approaches, the spatial imputation by SpatialPCA is also recommended to be performed only on points in the space that are surrounded by observed expression (lines 1200-1203 on page 28). Indeed, extrapolating outside the measured tissue area would be challenging for any methods, especially when the extrapolated area contains distinct cell type composition and transcriptomic profiles as compared to the measured area. We also agree with your comment on the density of spots in imputation. Indeed, likely most spatial imputation methods, the previous version of SpatialPCA requires the user to appropriately choose the density of spots they want to impute. We note, however, that some reasonable default density choices might be possible for certain spatial transcriptomics technologies. For example, the ST technology arrange spots on square while the Visium technology arrange spots on hexagonal lattices; both provide a natural way to define a neighborhood structure as squares or hexagons. Therefore, following your comments, we have followed BayesSpace and implemented the default density choices for these two technologies, where we impute nine subspots for each ST spot and impute six subspots for each Visium spot. For all other technologies, we simply set the default choice to be imputing on four new locations for each measured location. The new implementations are updated in our software and are described in the updated Method section (lines 1189-1191 on page 28).

(2) The output of the SpatialPCA method are latent factors which can be used as an input to downstream existing methods. The authors have paired their spatial PCs with three types of downstream methods to address the following questions:

- Spatial transcriptomic visualization
- this is done by
 1. Dim red via SpatialPCA,
 2. More dim red (umap or tsne) to three dimensions,
 3. assigning RGB to each of the three axes,
 4. Plot the colors.
- Spatial domain detection

- pair SpatialPCA with existing clustering method - the method seems to be good at detecting spatially contiguous and smoothly varying spatial domains - which is not always a realistic assumption, for example in the presence of a stripe pattern.

Thank you for the nice summary and the last comment on the stripe pattern. Following your comment, we have introduced additional simulations with a stripe pattern to examine the performance of SpatialPCA and the other methods in detecting spatially non-contiguous and non-smoothly varying domains (Supplementary Figure S4F). In particular, we divided the tissue into six equal-sized stripes and denoted the three odd stripes as domain one and the three even stripes as domain two. We set the proportions of the four cell types to be different in the two domains: in domain one, the cell types with the highest to lowest proportions are cell types 1, 2, 3, and 4; in domain two, the four types with the highest to lowest proportions are cell types 2, 3, 4, and 1. We considered three cell type composition scenarios similar to those used in the original simulations. Specifically, in the first scenario, each layer contains two dominant cell types, with 60% of the cells belonging to the dominant cell types and 40% of the cells belong to the other cell types. In the second scenario, each spatial domain contains two major cell types with unequal proportion, with one consisting of 60% of cells and the other consisting of 30% of cells, along with two minor cell types each consisting of 5% of cells. In the third scenario, each spatial domain contains three major cell types, with one consisting of 60% of cells, along with two each consisting of 20% of cells. The details are described in detail in the updated Methods section (lines 824-838 on page 20).

In the new simulations, we found that SpatialPCA also outperforms the other methods in all three scenarios in detecting the stripe pattern. Specifically, in the first scenario, SpatialPCA achieves an ARI of 0.904, which is higher than the other methods (BayesSpace=0.164, SpaGCN=0.521, HMRF=0.859, NMF=0.04, PCA=0.067). In the second scenario, SpatialPCA achieves an ARI of 0.854, which is higher than the other methods (BayesSpace=0.063, SpaGCN=0.585, HMRF=0.707, NMF=0.094, PCA=0.034). In the third scenario, SpatialPCA achieves an ARI of 0.86, which is also higher than the other methods (BayesSpace=0, SpaGCN=0.523, HMRF=0.641, NMF=0.166, PCA=0.1). The new results are shown in Supplementary Figures S4 and are described in detail in the updated Result section (lines 200-204 on page 5).

- Spatial trajectory inference

- they pair SpatialPCA with an existing trajectory inference method. Single-cell RNA-seq trajectory inference methods are designed to be applied to single cell resolution data. It is unclear how reliable such methods are for larger/mixed spatial transcriptomics spots.

Thank you for the comment. Indeed, the single-cell RNA-seq trajectory inference methods were initially designed with single cell data in-mind. However, some recent studies have started to explore the use of single-cell RNA-seq trajectory inference methods for larger/mixed spatial transcriptomics spots. For example, ref [9] calculated spot level "Spacetime" in 10X Visium data using the single-cell trajectory inference method Slingshot. Ref [10] estimated the spot level pseudotime in 10X Visium using the single-cell trajectory inference methods PHATE [11] and

monocle [12]. stLearn [13] applied single-cell trajectory inference method PAGA [14] on spatially smoothed gene expression data at spot level. In the present study, we also found that pairing SpatialPCA with existing single-cell trajectory inference methods appear to be promising in the examined spatial transcriptomics datasets. Therefore, following your comments, we have cited these existing literatures on applying single-cell trajectory methods for spatial transcriptomics in the updated manuscript to support such use (line 1209 on page 28). Certainly, we do fully agree with you that the use of single-cell RNAseq trajectory inference methods in spatial transcriptomics is likely not optimal. Therefore, following your comment, we have also included a couple of sentences in the Discussion section to emphasize that future method development for trajectory inference that specifically targeted for larger/mixed spatial transcriptomics spots is an important direction and that pairing SpatialPCA with these future methods will likely have added benefits and improved accuracy for spatial trajectory inference (lines 576-582 on page 14).

(3) First, the performance of the SpatialPCA method is evaluated through simulations and comparisons to other methods. The quality of the factors derived from SpatialPCA is evaluated in the context of one of the downstream tasks – domain detection. The authors simulate a simplified layered tissue consisting of 4 layers, 10k single cell locations from 4 cell types, across 5000 genes. To match the lack of single-cell resolution of many spatial transcriptomics technologies, the authors perform spot aggregation at different resolutions. The counts for simulated cells are binned, and not ever “split” between neighboring spots, which is likely to happen in technologies such as Slide-seq and ST or Visium, for example. This sharing of counts among neighboring spots will lead to artifactually higher spatial correlation between spots in real data, and is not well represented in the simulation design.

Thank you for the comment. We originally binned the single cells into pseudo-spots following the standard binning strategy used in previous studies (e.g. [15]). The spatial correlation created in the simulations from our binning strategy appears to be consistent with what we have observed in the real datasets. Specifically, the Moran’s I value ranges from -0.001 to 0.286 (median=0.001) in the LIBD data; from -0.006 to 0.161 (median=0.004) in the ST tumor data; from -0.0007 to 0.003 (median=-0.0001) in the Slide-Seq data; and from -0.0005 to 0.058 (median=0.0004) in the Slide-Seq V2 data. In the simulations created by the binning strategy, the median Moran’s I value is 0.001, 0.001, and 0.002, when the spot diameter is 90um, 107um, and 145um, respectively. Therefore, the spatial correlation created in the previous simulations appear to match the real data reasonably well.

In addition, following your comment, we also searched the literature but were unable to find any previous studies that used the “split” strategy brought up in your comment. However, we have decided to follow your comment and introduced a new set of simulations using a split strategy, where we split the cells and introduce additional spatial correlation between spots. Specifically, we first obtained 10,000 cells and assigned them to the spatial locations in the same four scenarios described in the previous simulations. In the process, we randomly selected 2,500 cells to be split. For each of these cells, we randomly split its expression level into four components based on uniform random weights that sum up to one. We then added each of the four

components randomly to the expression level of one of its four nearest neighboring cells. This way, we obtain a total of 7,500 pseudo-cells, 86.5% across simulation replicates contain split expression level from some of its neighboring cells. Afterwards, we aggregate the 7,500 pseudo-cells into spots. With this cell split strategy, the nearby spots on the tissue tend to have more similar expression profiles than the original binning strategy, thus introducing higher spatial correlation between spots. Indeed, the median spatial correlation now increases to 0.003, 0.003, and 0.005, when the spot diameter is 90um, 107um, and 145um, respectively. The new simulations are described in detail in the updated Method section (lines 840-851 on page 20).

In the new simulations, we found that the performance of all methods improved due to the increased spatial correlation, which makes it easier to identify spatial domains. The performance of SpatialPCA remains either the highest or very close to the highest among the methods. For example, when spot diameter is 90um, the median ARI value achieved by SpatialPCA across simulation replicates is 0.954, 0.952, 0.967, and 0.963 for the four scenarios, respectively. SpaGCN works well in scenario III when minor cell types have lower proportion in each spatial domain, but its performance decays in other scenarios when proportion of minor cell types increase in each spatial domain (median ARI=0.933, 0.936, 0.966, and 0.913 in four scenarios). HMRF performs worse than SpaGCN in all scenarios (median ARI=0.826, 0.807, 0.905, and 0.853 in four scenarios). BayesSpace does not perform as well as SpatialPCA or SpaGCN (median ARI=0.839, 0.838, 0.905, 0.839 for the four scenarios). While PCA and NMF perform the worst (PCA median ARI=0.579, 0.555, 0.77, and 0.602; NMF median ARI=0.625, 0.614, 0.791, and 0.651 in the four scenarios). The new results are shown in Supplementary Figures S5 and are described in the updated Results section (lines 204-208 on page 5).

(4) Finally, the authors apply their method, SpatialPCA, paired with existing downstream analysis methods, to four public spatial transcriptomics datasets from different technologies and different tissue structures: cortex, mouse hippocampus, and tumor microenvironment.

One of the methods chosen as a benchmark is NMF, i.e. Non-negative matrix factorization. As the name implies, the method is designed to be applied to non-negative data. However, the authors apply it to data after normalization as described in the Methods sections, when the data is no longer non-negative. This makes the comparisons to NMF inappropriate.

Thank you for the comment. We previously followed [16] and used SCTransform function to normalize the gene expression data and further converted all negative values to zero before performing NMF. Therefore, NMF was applied to non-negative data. We apologize for not making these details apparent, which were described in the previous Methods section. Following your comment, we have performed new analyses using a more common data normalization approach for NMF: we first performed log normalization on the gene expression count matrix using logNormCounts function in the scater package following the tutorial on <https://rdrr.io/bioc/scater/man/runNMF.html>, and then used the calculateNMF function to obtain the PCs from NMF implemented in scater [17] package. Compared to the previous data transformation strategy for NMF, the updated NMF has slightly decreased clustering accuracy

but better spatial continuity. For example, in the DLPFC data, the median ARI reduced by 0.1, and the median PAS score also reduced by 0.06. The updated results are shown in Figure 1-5, Supplementary Figures S1-S5, S11-S12, S19, S22-S23, S26, S29, S35-S37, S39 and S43, Supplementary Tables S1-S5, S7 and S13, and are described in detail in the updated Results (line 251 on page 6, lines 263 and 269 on page 7, lines 314-315 on page 8, lines 384-385 on page 9, line 429 on page 10, lines 440, 448, 457 and 459 on page 11) and Method section (lines 920-922 and lines 934-372 on page 22).

(5) A key metric used by the authors in the evaluation of the ability of SpatialPCA to predict pathologist annotations is McFadden's pseudo R^2 . This metric is not described in the methods and it remains unclear to this reviewer how it can be applied to PCA or NMF, as they are not identified with a likelihood.

Thank you for the comment. These details were previously described in the "Spatial clustering performance evaluation" subsection in the Methods section. "Specifically, we treated the true spatial domains as the outcome and fitted a multinomial regression model. For the dimension reduction methods, we treated the extracted low-dimensional components as predictors in the regression model. In the regression model, we computed the McFadden adjusted pseudo- R^2 [18] to evaluate the predictive ability of the predictor variables in predicting the ground truth. A higher pseudo- R^2 suggests that the method is capable of extracting informative output in predicting the true spatial domains." (current lines 933-938 on page 22). Following your comment, we have included a sentence in the Results section (lines 247-248 on page 6) where we first introduced pseudo R^2 to point readers to these important details.

(6) The method is reported to not be sensitive to choice of hyperparameters, such as number of factors. However, the evaluation is purely visual and subjective (see Figures S5, S10, S14, S19).

Thank you for the comment. Following your comment, we have included additional quantitative measurements for the sensitivity analyses. Specifically, we included the comparison on ARI, NMI, PAS, and CHAOS scores for the DLPFC and ST tumor datasets where ground truth domain annotations are available. We have included comparisons on PAS and CHAOS scores for the Slide-seq and Slide-seq V2 data where ground truth domain annotations are unavailable. Consistent with previous visual inspections, SpatialPCA performs robustly well across different sensitivity settings in different datasets. The updated results are shown in the updated Supplementary Figures S9, S20, S27, and S34 and are described in the updated Results section (lines 256-258 on page 7, lines 307-309 on page 8, lines 376-378 on page 9, lines 441-443 on page 11).

(7) The work has potential to be useful to researchers seeking to identify spatial domains de novo. It should be pointed out that the work bares a large degree of similarity with:
<https://doi.org/10.48550/arXiv.2110.06122> Nonnegative spatial factorization by F. William Townes, Barbara E. Engelhardt

<https://doi.org/10.1038/s41592-021-01343-9> Identifying temporal and spatial patterns of variation from multimodal data using MEFISTO by Velten et al.

Thank you. We have included both references to the updated manuscript. (We did not cite these two papers in the previous version because these two papers have not been posted when our manuscript was initially submitted to and under revised in a different journal almost two years ago.)

Reviewer #3 (Remarks to the Author): Expert in bioinformatics, cancer genomics, immunology, and tumour microenvironment

In this study, the authors present a dimensionality reduction approach tailored for spatial transcriptomics. The authors show the new methods utility across simulations and multiple datasets and compared to other methods. Altogether, this is an impressive piece of work in an up-and-coming field of research. The method seems sound and there is a lot of work here showing its usability to extract insights.

I focus here on comments on the HER2 tumor data analysis

Thank you for your constructive comments. Our detailed responses are listed below.

1. The original paper provides data on 36 sections, and there are manual annotations in 6 of them. The analysis shown here focuses on one sample only, and shows marginal advantage for SpatialPCA over other methods, and can be random. I think that it would be much more powerful to show statistics across all samples.

Thank you for the comment. The original data paper provided spatial domain annotation for 8 tissue sections (A1, B1, C1, D1, E1, F1, G2 and H1), among which H1 and G2 contain the TLS region. Because H1 has a higher number of spatial locations (n=613) than G2 (n=467), we used it as the main example in the original manuscript.

Following your comments, we have included additional analysis on all eight annotated samples in the updated manuscript. We did not compare methods in the other 24 tissue sections because of a lack of annotations for those sections. The new comparative results are similar to what we have observed on the H1 sample. Briefly, SpatialPCA achieves the highest median ARI and NMI scores among all compared methods across all eight tissue sections. SpatialPCA achieves similar median PAS and CHAOS scores as BayesSpace, both of which have lower scores than the other methods, across eight tissue sections. The new results are shown in the updated Supplementary Figures S36 and are described in detail in the updated Results section (lines 448-451 on page 11).

2. From the text it is suggested that there is very high matching between the expert annotation and the annotation by SpatialPCA, but by eye it doesn't look that well. For example, DCIS, cancer and fibrous tissue are not distinguished at all. There are many differences. I understand that other methods aren't doing better, but this suggests to me that this whole benchmarking approach – trying to identify major domains – might not make much sense. The domains identified by the pathologist are very “high-level” and there shouldn't be much resemblance to the fine-grained analysis that is possible with ST.

Thank you for the comment. We fully agree. Indeed, currently there is an unfortunate lack of fine-grained annotations by pathologists in the field of spatial transcriptomics, which impedes

method comparison at a fine scale. Consequently, most existing studies had to rely on the available high-level annotations to evaluate and compare method performance (e.g. [19, 20]), which will inevitably miss the fine-grained details that are contained in the collected ST datasets. While it is practically infeasible to obtain fine-scaled annotations in the analyzed ST datasets, we decided to follow the essence of your comment and performed a set of fine-grained analysis in the DCIS cancer dataset. Specifically, we reasoned that gene expression would capture fine-grained details for the spatial domains on the tissue much better than the high-level annotations can. Therefore, if we were able to obtain a list of genes that are specifically expression in a spatial domain, then their gene expression on the tissue will likely define the boundary of the spatial domain with fine-grained details. To do so, for each spatial domain in turn, we first performed differential expression (DE) analysis using MAST on the histologist annotated high-level annotations to obtain a list of domain-specific DE genes (# of DE genes ranges from 19 to 408 for different domains). Next, for each spatial domain in turn, we obtained the averaged expression level of its domain-specific DE genes to serve as the domain-specific meta-gene. The expression of the meta-gene for each domain likely contains the fine-grained transcriptomic information and can thus be used to evaluate the fine-grained performance of different spatial domain detection methods. Therefore, we examined the enrichment of each of the seven meta-genes in the spatial domains detected by each method for method evaluation. Intuitively, if the spatial domain detection method is better powered to detect fine-grained spatial domains, then the expression level of the meta-genes would be enriched in one of the detected spatial domains. We calculated an enrichment score as the ratio of the meta-gene expression in the corresponding domain vs the other domains. The analysis details are provided in the updated Methods section (lines 1079-1105 on pages 25-26). As expected, SpatialPCA achieves the highest meta-gene enrichment score (1.417) compared with the other methods (BayesSpace 1.269; HMRF 1.202; NMF 1.345; PCA 1.248; SpaGCN 1.096) on sample H1. SpatialPCA also achieves the highest meta-gene enrichment score (1.141) compared with the other methods (BayesSpace 1.082; HMRF 0.916; NMF 0.983; PCA 0; SpaGCN 0.822) across all eight annotated samples. The new results support that the spatial domains detected by SpatialPCA likely captured the fine-grained transcriptomic architecture on the tissue, at least more so than the other methods. The new results are shown in the updated Supplementary Figures S36B-S36D and are described in detail in the updated Results section (lines 451-459 on page 11).

3. “We found that the regional specific genes detected by SpatialPCA are highly enriched in pathway of immune response and biological adhesion” – this sentence is unclear to me.

Thank you. To clarify the sentence, we have revised it to read “We found that the DE genes in the immune region detected by SpatialPCA are highly enriched in the pathways of immune response, while the DE genes in the tumor region detected by SpatialPCA are highly enriched in the pathways of biological adhesion” (lines 468-470 on page 11).

4. I don’t understand that deconvolution analysis that included only immune cells. This surely creates bias. For example, the mast cell result is most probably some kind of bias (to my best

knowledge, there aren't supposed to be many mast cells). My guess is that since the reference of mast cells have some resemblance to stroma cells, it is "enriched" in this analysis.

Thank you for the comment. We previously performed two different types of deconvolution analyses: one using only the immune cells while the other using both immune cells and malignant cells. The first type of deconvolution analysis primarily follows the standard practice in the field of performing deconvolution of the tumor tissue using only immune cells (e.g. following CIBERSORT[21] and quanTIseq[22]). Using immune cells for deconvolution allows us to focus on examining the composition and density of immune cells in the tumor microenvironment, which is a critical indicator of tumor growth, cancer progression, and the success of anticancer therapies (e.g. [22]). We fully agree with your comment that bias, especially bias of mast cells, may occur in the first type of deconvolution analysis. Because of this potential bias, we previously also performed the second type of deconvolution analysis using a range of cell types beyond immune cells, which yields qualitatively similar results as the first type of deconvolution analysis. Following your comment, we removed the first type of deconvolution analysis, which does not alter the conclusion of the paper.

5. "In PCA and NMF, the detected tumor regions are not as smooth and continuous as SpatialPCA." – of course, SpatialPCA underlying hypothesis is of smoothness, and the method inherently pushes for that.

Thank you for the comment. We fully agree. To make this explicit to readers, we have modified this sentence to "In PCA and NMF, the detected tumor regions are not as smooth and continuous as SpatialPCA, as the later explicitly models spatial correlation in dimension reduction." (lines 475-477 on page 11).

We also note that, while the spatial smoothness assumption is not employed in dimension reduction methods PCA and NMF, such spatial smoothness assumption is commonly employed for almost all spatial domain detection methods. For example, BayesSpace uses a Potts model to borrow the information from nearby spots to achieve spatial smoothing. SpaGCN induces spatial smoothness by aggregating spatial locations and histology information into a graph convolution layer. stLearn uses a morphological normalization step to incorporate spatial smoothness into the gene expression. HMRF builds a graph to represent the spatial relationship among the cells and detects spatial domain by comparing the gene expression of each cell with its surroundings to search for coherent spatial patterns. Therefore, while the detected tumor regions from SpatialPCA are expected to be smoother and more continuous than that from PCA and NMF, such *a priori* expectation does not hold for comparing SpatialPCA with the other methods.

6. "with multiple unique features of TLS: the region is located near the tumor; the region primarily contains T cells and B cells, all of which are key cell types in TLS" – those are not unique features of TLS, but for almost any region enriched with immune cells in the TME. I am not an expert, but there are more markers required to determine a region as TLS. See Box 1 here

<https://www.nature.com/articles/s41568-019-0144-6>. Since it is possible to identify TLS with H&E, I think it would be best to compare predictions of TLS to an expert determination.

Thank you for the comment. To address your comment, (1) we have removed “unique” in the sentence following your suggestion. (2) We note that we actually previously relied on the TLS marker genes listed in Box 1 in your referred paper <https://www.nature.com/articles/s41568-019-0144-6> and selected those that are marker genes for breast cancer and those that are expressed in our data for the analysis (e.g. Figure 5). We now make this important point clear in the updated Results section (lines 483-484 on page 11). (3) In addition, we have performed additional analysis by obtaining the TLS score for each tissue location from the original study [23]. The TLS score was computed in the original study based on the interaction strength between B cells and T cells in each spatial location, which is a key feature of TLS. We indeed found that the TLS scores are highly enriched in the TLS region detected by SpatialPCA (Supplementary Figure S40). The new results are shown in the updated Supplementary Figures S36B-S36D and are described in detail in the updated Results section (lines 484-487 on pages 11-12).

7. The identification of the surrounding region is interesting, but I think there are too many speculations here for a results section, and most of the text here should be moved to the discussion.

Thank you for the comment. Following your comment, we have moved part of the results on tumor surrounding region detection and interpretation to the Discussion section (lines 542-545 on page 13). We kept the other part of the results on tumor surrounding region in the Results section in order to address your comment #6.

Minor comments:

BayesSpace seems to be better in some of the statistics (PAS and CHAOS). It should be noted.

Thank you. We have noted in the Results section that BayesSpace is better in some statistics in some datasets (e.g. lines 444-445 on page 11).

Supp figure are not in order (S30, S29)

Figure S18 –

a. Missing axes

b. Measuring variance doesn't seem like a fair approach. SpatialPCA is actively smoothing the data, so variance is expected to be low.

c. Not clear what is the first row.

d. Why no comparison to BayesSpace and SpaGCN

5f-h before 5e

Thank you. We have fixed these issues in the updated manuscript. For b, this is actually exactly the point we want to make. By modeling spatial correlation and smoothing the data in a spatial fashion, SpatialPCA will produce lower variance. We added a sentence in the figure legend to make this point clear. For Figure S18 (current Figure S32), we did not compare to BayesSpace and SpaGCN in the RGB plot in (d) because RGB plot is used for visualizing low dimensional components, but these two methods are not dimension reduction methods and thus do not produce low dimensional components.

References:

1. Ruben Dries QZ, Rui Dong, Chee-Huat Linus Eng, Huipeng Li, Kan Liu, Yuntian Fu, Tianxiao Zhao, Arpan Sarkar, Feng Bao, Rani E George, Nico Pierson, Long Cai, Guo-Cheng Yuan. Giotto, a toolbox for integrative analysis and visualization of spatial expression data. *bioRxiv*. 2020;701680. doi: <https://doi.org/10.1101/701680>.
2. Moffitt JR, Bambah-Mukku D, Eichhorn SW, Vaughn E, Shekhar K, Perez JD, et al. Molecular, spatial, and functional single-cell profiling of the hypothalamic preoptic region. *Science*. 2018;362(6416). Epub 2018/11/06. doi: 10.1126/science.aau5324. PubMed PMID: 30385464; PubMed Central PMCID: PMC6482113.
3. Li Z, Zhou X. BASS: multi-scale and multi-sample analysis enables accurate cell type clustering and spatial domain detection in spatial transcriptomic studies. *Genome Biology*. 2022;23(1):168. doi: 10.1186/s13059-022-02734-7.
4. Senechal M. Spatial Tessellations - Concepts and Applications of Voronoi Diagrams - Okabe,a, Boots,B, Sugihara,K. *Science*. 1993;260(5111):1170-&. doi: DOI 10.1126/science.260.5111.1170. PubMed PMID: WOS:A1993LC94200046.
5. Street K, Risso D, Fletcher RB, Das D, Ngai J, Yosef N, et al. Slingshot: cell lineage and pseudotime inference for single-cell transcriptomics. *BMC Genomics*. 2018;19(1):477. Epub 2018/06/20. doi: 10.1186/s12864-018-4772-0. PubMed PMID: 29914354; PubMed Central PMCID: PMC6007078.
6. Ji Z, Ji H. TSCAN: Pseudo-time reconstruction and evaluation in single-cell RNA-seq analysis. *Nucleic Acids Res*. 2016;44(13):e117. Epub 2016/05/15. doi: 10.1093/nar/gkw430. PubMed PMID: 27179027; PubMed Central PMCID: PMC4994863.
7. Shin J, Berg DA, Zhu Y, Shin JY, Song J, Bonaguidi MA, et al. Single-Cell RNA-Seq with Waterfall Reveals Molecular Cascades underlying Adult Neurogenesis. *Cell Stem Cell*. 2015;17(3):360-72. Epub 2015/08/25. doi: 10.1016/j.stem.2015.07.013. PubMed PMID: 26299571; PubMed Central PMCID: PMC48638014.
8. Setty M, Tadmor MD, Reich-Zeliger S, Angel O, Salame TM, Kathail P, et al. Wishbone identifies bifurcating developmental trajectories from single-cell data. *Nat Biotechnol*. 2016;34(6):637-45. Epub 2016/05/03. doi: 10.1038/nbt.3569. PubMed PMID: 27136076; PubMed Central PMCID: PMC4900897.
9. Tower RJ, Li Z, Cheng YH, Wang XW, Rajbhandari L, Zhang Q, et al. Spatial transcriptomics reveals a role for sensory nerves in preserving cranial suture patency through modulation of BMP/TGF-beta signaling. *Proc Natl Acad Sci U S A*. 2021;118(42). Epub 2021/10/20. doi: 10.1073/pnas.2103087118. PubMed PMID: 34663698; PubMed Central PMCID: PMC8545472.
10. Mantri M, Scuderi GJ, Abedini-Nassab R, Wang MFZ, McKellar D, Shi H, et al. Spatiotemporal single-cell RNA sequencing of developing chicken hearts identifies interplay between cellular differentiation and morphogenesis. *Nat Commun*. 2021;12(1):1771. Epub 2021/03/21. doi: 10.1038/s41467-021-21892-z. PubMed PMID: 33741943; PubMed Central PMCID: PMC7979764.
11. Moon KR, van Dijk D, Wang Z, Gigante S, Burkhardt DB, Chen WS, et al. Visualizing structure and transitions in high-dimensional biological data. *Nat Biotechnol*. 2019;37(12):1482-92. Epub 2019/12/05. doi: 10.1038/s41587-019-0336-3. PubMed PMID: 31796933; PubMed Central PMCID: PMC7073148.

12. Trapnell C, Cacchiarelli D, Grimsby J, Pokharel P, Li S, Morse M, et al. The dynamics and regulators of cell fate decisions are revealed by pseudotemporal ordering of single cells. *Nat Biotechnol.* 2014;32(4):381-6. Epub 2014/03/25. doi: 10.1038/nbt.2859. PubMed PMID: 24658644; PubMed Central PMCID: PMC4122333.
13. Duy Pham XT, Jun Xu, Laura F. Grice, Pui Yeng Lam, Arti Raghobar, Jana Vukovic, Marc J. Ruitenberg, Quan Nguyen. stLearn: integrating spatial location, tissue morphology and gene expression to find cell types, cell-cell interactions and spatial trajectories within undissociated tissues. *bioRxiv.* 2020. doi: <https://doi.org/10.1101/2020.05.31.125658>.
14. Wolf FA, Hamey FK, Plass M, Solana J, Dahlin JS, Gottgens B, et al. PAGA: graph abstraction reconciles clustering with trajectory inference through a topology preserving map of single cells. *Genome Biol.* 2019;20(1):59. Epub 2019/03/21. doi: 10.1186/s13059-019-1663-x. PubMed PMID: 30890159; PubMed Central PMCID: PMC6425583.
15. Stickels RR, Murray E, Kumar P, Li JL, Marshall JL, Di Bella DJ, et al. Highly sensitive spatial transcriptomics at near-cellular resolution with Slide-seqV2. *Nature Biotechnology.* 2021;39(3):313-9. doi: 10.1038/s41587-020-0739-1. PubMed PMID: WOS:000598987100001.
16. Lee Y, Bogdanoff D, Wang Y, Hartoularos GC, Woo JM, Mowery CT, et al. XYSeq: Spatially resolved single-cell RNA sequencing reveals expression heterogeneity in the tumor microenvironment. *Sci Adv.* 2021;7(17). Epub 2021/04/23. doi: 10.1126/sciadv.abg4755. PubMed PMID: 33883145; PubMed Central PMCID: PMC8059935.
17. McCarthy DJ, Campbell KR, Lun AT, Wills QF. Scater: pre-processing, quality control, normalization and visualization of single-cell RNA-seq data in R. *Bioinformatics.* 2017;33(8):1179-86. Epub 2017/01/16. doi: 10.1093/bioinformatics/btw777. PubMed PMID: 28088763; PubMed Central PMCID: PMC5408845.
18. McFadden D, University of California Berkeley. Institute of Urban & Regional Development. Conditional logit analysis of qualitative choice behavior. Berkeley, Calif.: Institute of Urban and Regional Development, University of California; 1973. 48 leaves p.
19. Hildebrandt F, Andersson A, Saarenmaa S, Larsson L, Van Hul N, Kanatani S, et al. Spatial Transcriptomics to define transcriptional patterns of zonation and structural components in the mouse liver. *Nat Commun.* 2021;12(1):7046. Epub 2021/12/04. doi: 10.1038/s41467-021-27354-w. PubMed PMID: 34857782; PubMed Central PMCID: PMC8640072.
20. Zhao E, Stone MR, Ren X, Guenthoer J, Smythe KS, Pulliam T, et al. Spatial transcriptomics at subspot resolution with BayesSpace. *Nat Biotechnol.* 2021;39(11):1375-84. Epub 2021/06/05. doi: 10.1038/s41587-021-00935-2. PubMed PMID: 34083791; PubMed Central PMCID: PMC8763026.
21. Newman AM, Liu CL, Green MR, Gentles AJ, Feng W, Xu Y, et al. Robust enumeration of cell subsets from tissue expression profiles. *Nat Methods.* 2015;12(5):453-7. Epub 2015/03/31. doi: 10.1038/nmeth.3337. PubMed PMID: 25822800; PubMed Central PMCID: PMC4739640.
22. Finotello F, Mayer C, Plattner C, Laschober G, Rieder D, Hackl H, et al. Molecular and pharmacological modulators of the tumor immune contexture revealed by deconvolution of RNA-seq data. *Genome Med.* 2019;11(1):34. Epub 2019/05/28. doi: 10.1186/s13073-019-0638-6. PubMed PMID: 31126321; PubMed Central PMCID: PMC6534875.
23. Andersson A, Larsson L, Stenbeck L, Salmen F, Ehinger A, Wu SZ, et al. Spatial deconvolution of HER2-positive breast cancer delineates tumor-associated cell type interactions.

Nat Commun. 2021;12(1):6012. Epub 2021/10/16. doi: 10.1038/s41467-021-26271-2. PubMed
PMID: 34650042; PubMed Central PMCID: PMC8516894.

REVIEWERS' COMMENTS

Reviewer #1 (Remarks to the Author):

I thank the authors for their careful and detailed work in addressing my comments. The revised manuscript is much improved with a methodological extension to multiple distinct spatial samples, sensitivity analysis with gamma parameter choice, demonstration of applicability to spatial FISH data, and added functionality to non-euclidean distances. I have no further concerns.

NOTE FROM THE EDITOR: Reviewer #1 considered that the concerns from Reviewer #2 were addressed.

Reviewer #3 (Remarks to the Author):

The authors' response to my concerns is satisfactory.

One issue that bothers me is regarding reviewer 1's comment about the gamma parameter. I see the response, however, it seems that the range used for the parameter (0.02 to 0.05) is quite specific. I think it would be worth expanding this range to understand how it changes the results. Actually, the ARI changed quite considerably for this small range (0.54-0.646), suggesting that the model isn't robust for this change.

Reviewer #3 (Remarks to the Author):

The authors' response to my concerns is satisfactory.

One issue that bothers me is regarding reviewer 1's comment about the gamma parameter. I see the response, however, it seems that the range used for the parameter (0.02 to 0.05) is quite specific. I think it would be worth expanding this range to understand how it changes the results. Actually, the ARI changed quite considerably for this small range (0.54-0.646), suggesting that the model isn't robust for this change.

Thank you for the comment. We note that the ARI range of 0.54-0.646 is actually not large and does not affect our conclusion. As a comparison, we have varied the bandwidth parameter of SpaGCN in the same range (0.02 to 0.05) and the resulting ARI from SpaGCN varied from 0.29 to 0.48. Importantly, SpatialPCA outperforms SpaGCN in all 31 bandwidth values, with an average ARI improvement of 62%. In addition, we expanded the bandwidth range from 0.01 to 1 and found the ARI from SpatialPCA ranges from 0.49 to 0.65 while the ARI from SpaGCN ranges from 0.25 to 0.50. In these settings, SpatialPCA again outperforms SpaGCN in 52 out of these 55 bandwidth values, with an average ARI improvement of 46%. The new results are shown in Supplementary Figure S10A (attached here for convenience) and described in the corresponding figure legend.